# Tropical Atlantic Climate and Ecosystem Regime Shifts during the Paleocene-Eocene Thermal Maximum

Frieling, Joost[1], Reichart, Gert-Jan[2,3], Middelburg, Jack J.[2], Röhl, Ursula[4], Westerhold, Thomas[4],

Bohaty, Steven M.[5], and Sluijs, Appy[1]

[1] Marine Palynology and Paleoceanography, Laboratory of Palaeobotany and Palynology, Department of Earth Sciences, Faculty of Geosciences, Utrecht University, Heidelberglaan 2, 3584CS Utrecht, Netherlands

[2] Department of Earth Sciences, Faculty of Geosciences, Utrecht University, Princetonplein 9, 3584CC Utrecht, Netherlands

[3] NIOZ Royal Netherlands Institute for Sea Research, 1790AB Den Burg, Texel, Netherlands

[4] MARUM - Center for Marine Environmental Sciences, University of Bremen, Leobener Straße 8, 28359, Bremen, Germany

[5] Ocean and Earth Science, National Oceanography Centre Southampton, University of Southampton, Waterfront Campus European Way Southampton, SO14 3ZH United Kingdom

*Correspondence to*: J. Frieling (j.frieling1@uu.nl)

**Abstract.** The Paleocene – Eocene Thermal Maximum (PETM; 56 Ma) was a phase of rapid global warming associated with massive carbon input into the ocean-atmosphere system from a [13]C-depleted reservoir. Many mid- and high-latitude sections have been studied and document changes in salinity, hydrology and sedimentation, deoxygenation, biotic overturning and migrations, but detailed records from tropical regions are lacking. Here, we study the PETM at Ocean Drilling Program (ODP) Site 959 in the equatorial Atlantic using a range of organic and inorganic proxies and couple these with dinoflagellate cyst (dinocyst) assemblage analysis. The PETM at Site 959 was previously found to be marked by a ~3.8‰ negative carbon isotope excursion (CIE), and a ~4 ºC surface ocean warming from the uppermost Paleocene to peak PETM, of which ~1 ºC occurs before the onset of the CIE. We record upper Paleocene dinocyst assemblages that are similar to PETM assemblages as found in extra-tropical regions, confirming poleward migrations of ecosystems during the PETM. The early stages of the PETM are marked by a typical acme of the tropical genus *Apectodinium*, which reaches abundances of up to 95%. Subsequently, dinocyst abundances diminish greatly, as do carbonate and pyritized silicate microfossils. The combined paleoenvironmental information from Site 959 and a close by shelf site in Nigeria implies the general absence of eukaryotic surface-dwelling microplankton during peak PETM warmth in the eastern equatorial Atlantic, most likely caused by heat stress. We hypothesize, based on a literature survey, that heat stress might have reduced calcification in more tropical regions, potentially contributing to reduced deep sea carbonate accumulation rates, and, by buffering acidification, also to biological carbonate compensation of the injected carbon during the PETM. Crucially, abundant organic benthic foraminiferal linings imply sustained export production, likely driven by prokaryotes. In sharp contrast, the recovery of the

CIE yields rapid (<<10 kyr) fluctuations in the abundance of several dinocyst groups, suggesting extreme ecosystem and environmental variability.

## 1. Introduction

Long-term gradual warming during the Late Paleocene and Early Eocene (~59-53 Ma) is recorded in the deep ocean (Zachos et al., 2008; Littler et al., 2014), as well as Southern hemisphere (Bijl et al., 2009; Hollis et al., 2012) and Northern hemisphere (Frieling et al., 2014) mid- and high-latitude surface oceans and equatorial Atlantic (Cramwinckel et al., in prep.). Superimposed, the Paleocene-Eocene Thermal Maximum (PETM) represents a transient period of rapid global warming, associated with the massive input of strongly [13]C-depleted carbon into the ocean-atmosphere system (Dickens et

al., 1995; Zeebe et al., 2009). This results in a ~2.5-8 ‰ negative carbon isotope excursion (CIE) in carbon-bearing substrates deposited during the PETM (McInerney and Wing, 2011; Sluijs and Dickens, 2012). The CIE has a distinct shape; a rapid "onset" (1-5 kyr; (Kirtland Turner and Ridgwell, 2016; Zeebe et al., 2016) followed by a prolonged (50-100 kyr) period, the "body", of stable low [13]C values and a recovery that lasts 42 - 82 kyr (Röhl et al., 2007; Abdul Aziz et al., 2008; Murphy et al., 2010; Westerhold et al., 2017) to values that remain slightly [13]C-depleted (0.5-1 ‰) relative to the latest

Paleocene. This pattern is best explained by massive (>>1500 Gt) carbon input from at least 1 but likely multiple reservoirs in the shape of $CO_2$ and/or $CH_4$; (Dickens et al., 1995; Panchuk et al., 2008; Zeebe et al., 2009; Dickens, 2011; Zeebe, 2013; Frieling et al., 2016; Luo et al., 2016).

The integrated changes in climate and the carbon cycle lead to global average surface warming during the PETM in the order of ~4-5 ºC (Dunkley Jones et al., 2013; Frieling et al., 2017), although this warming was not equally distributed across the

globe. Regions of enhanced warming are recognized in both the Northern mid and Southern mid and high latitudes, which shows the mechanism underlying extra-tropical amplification was not fully saturated even in these strong greenhouse climates (Frieling et al., 2017). The warming during the PETM is associated with sea level rise (Sluijs et al., 2008b), local photic zone euxinia (Sluijs et al., 2006, 2014, Frieling et al., 2014, 2017), global expansion of anoxic waters (Dickson et al., 2012) and an accelerated hydrological cycle (Pagani et al., 2006; Schmitz and Pujalte, 2007; Sluijs and Brinkhuis, 2009;

Handley et al., 2012; Carmichael et al., 2017).

The magnitude of warming during the PETM and environmental and ecological effects have been extensively documented at mid and high-latitude sections (see review in Sluijs et al., 2014). Although micropaleontological studies indicate assemblage shifts and environmental perturbation in several tropical open ocean regions (e.g., Ocean Drilling Program (ODP) Site 865; Kelly et al., 1996, ODP Site 1001; Bralower et al., 1997) the underlying mechanisms remain unclear because causal relations

with physicochemical parameters are difficult to establish. Evidence from Tanzania, Nigeria and ODP Site 865 indicates that tropical surface oceans warmed by ~3 °C (Frieling et al., 2017). In the modern ocean, a relatively modest (0.5-1 ºC) warming already negatively affects biodiversity in tropical regions (Pandolfi et al., 2011). In the Anthropocene migrations to higher

latitudes are noted in a multitude of biota (Doney et al., 2012), similar to the PETM (Kelly et al., 1996; Crouch et al., 2001; Wing et al., 2005; Sluijs et al., 2007b), notably in the (sub)tropical dinoflagellate cyst (dinocyst) *Apectodinium*.

Recently, a massive decrease in abundance and diversity of dinocysts was found during the PETM in Nigeria, which was attributed to heat stress (Frieling et al., 2017). The impact and geographical extent of this heat-stress driven biodiversity drop however remain unknown, although similar heat-stress effects on marine biota may have been widespread in tropical regions (Aze et al., 2014; Yamaguchi and Norris, 2015). Although other stressors like stratification, salinity fluctuations and acidification may play a role, it should be noted that dinoflagellates typically thrive under such conditions, including high $CO_2$ (Hoins et al., 2015). Here, we analyze the PETM from ODP Site 959 in the equatorial Atlantic (Figure 1) as recently described using carbon isotope stratigraphy, biostratigraphy and $TEX_{86}$ paleothermometry (Frieling et al., submitted). Site 959 is located near the top of the Cote d'Ivoire – Ghana Transform Margin (CIGTM), a submarine high ~150 km offshore Ivory Coast (Shipboard Scientific Party, 1996). Upper Paleocene and lower Eocene sediments were deposited at a paleolatitude of 3 - 7º S (Seton et al., 2012; van Hinsbergen et al., 2015) and are typically composed of roughly equal amounts of carbonate, biogenic silicate and siliciclastics (Shipboard Scientific Party, 1996; Wagner, 2002) with some (0.1-1%) organic matter. We measure bulk magnetic susceptibility and apply inorganic proxies, including X-ray fluorescence core scanning and bulk sediment chemistry and combine these with dinocyst assemblages and previously published $TEX_{86}$, Branched versus Isoprenoid Tetraether (BIT) index data (Frieling et al., submitted) to reconstruct environments across the PETM.

## 2. Material

The ODP recovered a relatively complete Cretaceous and Cenozoic sediment sequence from Hole 959D, located on the CIGTM, in 1995 (3°37.656'N, 2°44.149'W; Figure 1). From the Early Cretaceous onwards, the submarine high has been subsiding, reaching bathyal depths around the Cretaceous – Paleogene boundary (Oboh-Ikuenobe et al., 1997); the present day water depth is ~2000m (Shipboard Scientific Party, 1996). Calcareous nannofossil biostratigraphy indicates the presence of the upper Paleocene CP8a-CP8b zone boundary in the interval from Core section 41R-6 to Core section 42R-2 mbs (Shafik et al., 1998). The PETM was identified in Core section 42R-1 and spans the interval 804.1 – 802.6 meters below sea floor (mbsf) based on the identification of a ~4‰ negative CIE and ~4 °C warming (Frieling et al., submitted)(Figure 2). Unfortunately, the top of the CIE lies within the ~1 m gap between Cores 41R and 42R. Notable lithological changes occur in the interval from 804.1 mbsf to ~803.8 mbsf, where dominantly carbonate and siliciclastic sedimentation is for a substantial part replaced by diagenetically altered biogenic silicates (porcellanite) (Shipboard Scientific Party, 1996; Wagner, 2002). The sediments above section 41R-6 are recognized by more frequent occurrences of porcellanite (Shipboard Scientific Party, 1996; Wagner, 2002). Sediments below Core section 42R-1 show apparently cyclic dark-light alterations, representing variable amounts of clay, carbonate and biogenic silica (Shipboard Scientific Party, 1996; Wagner, 2002). The position of the site, near the top of a submarine high, explains the relatively sparse siliciclastic supply to the core location.

We continuously sampled sections 41R-6, 41R-CC, 42R-1 and 42R-2 at the Bremen Core Repository (BCR) of the International Ocean Discovery Program (IODP), representing the interval from 800.5 to 805.58 mbsf at a resolution of 1-3 cm (see Frieling et al., submitted).

Frieling et al (submitted) recorded a ~4‰ negative CIE in the interval from 804.1 to 803.8 mbsf (Figure 2). Notably, this interval is marked by organic-lean biogenic silica and siliciclastics, which are likely derived from an allochthonous, potentially turbiditic, deposit, as reflected by anomalous Ti/Al ratios (Frieling et al., submitted) (see also Figure 3h). Carbon isotope stratigraphy of total organic carbon ($\delta^{13}C_{TOC}$) indicates the presence of a 2 ‰ negative step at the base of the porcellanite (804.1 mbsf), interpreted as the onset of the CIE, and stable low (~ -30‰) values from 803.8 to 803 mbsf, interpreted as the body phase of the CIE (Frieling et al., submitted, Figure 2a). The gradual decrease from ~ -27‰ at 804.09 mbsf to -30‰ at 803.8 mbsf is interpreted as a mixing line between organic matter produced during the Paleocene and the PETM (Frieling et al., submitted). From 803 to 802.6 mbsf there is an increase to values ~1.5 ‰ lower than background Paleocene. A core gap 801.6 - 802.6 mbsf conceals the end of the recovery phase. Core 41R-6 and 41R-CC are characterized by earliest Eocene values, about 0.7 ‰ lower than latest Paleocene. Although no indications for gaps were observed during sampling, the two distinct steps in the $\delta^{13}C_{TOC}$ record may indicate very small hiatuses; one at 804.1 mbsf and one at 803.0 mbsf (Figure 2). Paleocene accumulation rates are in the order of ~1.3 cm kyr$^{-1}$, based on calcareous nannofossil biostratigraphy (Shafik et al., 1998) and a cyclostratigraphic age model based on variations in TOCwt% (Frieling et al., submitted). Accumulation rates of ~1 cm kyr$^{-1}$ are used for the body of the CIE, although it is important to note that this assumes the body is complete and between 70 and 100 kyr in duration.

## 3 Methods

All discrete samples taken from the working halves were freeze-dried and measured for bulk magnetic susceptibility before splitting them into fractions for palynology, inorganic and organic geochemical analysis.

### 3.1 Bulk Magnetic susceptibility

Freeze-dried samples were weighed and measured for bulk magnetic susceptibility on an MFK1-FA at the Paleomagnetism laboratory, Fort Hoofddijk, Utrecht University. Reproducibility was determined by replicate measurements and was always better than 1%.

### 3.2 Palynology

We processed and counted a total of 155 samples, using standard protocols used at the Laboratory of Palaeobotany and Palynology at Utrecht University (Sluijs et al., 2003). In brief, a spike of exote *Lycopodium clavatum* spores (n = 20848 ± 691) was added to 1-10 g of freeze-dried sample to allow for absolute quantitative analysis (Stockmarr, 1971). To dissolve carbonates, sediments were first treated with 10% HCl, after which supernatants were decanted. This was followed up by

two steps of 38-40% HF and 30% HCl to dissolve silicates. Samples were centrifuged and neutralized with tap water before sieving over 250 and 15 μm sieves to remove large and small particles, respectively. Subsequently, residues were concentrated in glycerine water and mounted on microscope slides using glycerine jelly. We followed dinocyst taxonomy of Fensome and Williams, (2004) and the paleoecological grouping of Sluijs and Brinkhuis, (2009). Unlike Sluijs and Brinkhuis, (2009), we separate Protoperidinioid dinocysts from the other Peridinioid types with hexagonal 2a archeopyles. Where possible, a minimum of 200 dinocysts was counted. For samples with very low numbers of dinocysts, slides were fully counted to a total of ~10-100 dinocysts. We estimated the abundance of pyritized remains of biogenic silica relative to organic particles in each sample. A full list of encountered dinocyst taxa is given in Supplementary Table 1. All materials are stored in the collection of the Laboratory of Palaeobotany and Palynology, Utrecht University.

**3.3 XRF Core Scanning.**

XRF Core Scanner data were collected every 1 cm down-core over a 1 $cm^2$ area with slit size of 10 mm using generator settings of 10 kV and 50kV, a current of 0.2 mA and 1.0 mA respectively, and a sampling time of 30 seconds directly at the split core surface of the archive halves of Core sections 41R-6, 42R-2 and lower part of 42R-1 with XRF Core Scanner III at the MARUM - University of Bremen. The split core surface was covered with a 4 μm SPEXCerti Prep Ultralene foil to avoid contamination of the XRF measurement unit and desiccation of the sediment. The dark-colored interval from 803.8 mbsf to 803 mbsf and directly overlying sediment in Core section 42R-1 was too fragmented for acquiring reliable core scanning measurements. The here reported data have been acquired by a Canberra X-PIPS Detector (X-PIPS SXP5C-200-1500 from Canberra) with 150eV X-ray resolution, the Canberra Digital Spectrum Analyzer DAS 1000 and an Oxford Instruments 100W Neptune X-ray tube with rhodium (Rh) target material. Raw data spectra were processed by the Analysis of X-ray spectra by Iterative Least square software (WIN AXIL) package from Canberra Eurisys.

**3.4 Bulk sediment chemistry from Inductively Coupled Plasma - Optimal Emission Spectroscopy (ICP-OES)**

Bulk sediment chemistry was determined on 28 samples in the interval that could not be used for XRF core scanning and 22 samples from intervals with XRF scanning data to assess consistency between the two methods. Approximately 125 mg of powdered freeze-dried sediment was dissolved in 2.5 ml HF (40 %) and 2.5 ml of $HClO_4/HNO_3$ mixture, in a closed Teflon bomb at 90 °C during one night. The acids were then evaporated at 160 °C and the resulting gel was subsequently redissolved in 1M $HNO_3$ at 90 °C during another night. Subsequently, total elemental concentrations were determined by ICP-OES (Perkin Elmer Optima 3000 Inductively Coupled Plasma - Optimal Emission Spectroscopy) at Utrecht University. Precision and accuracy was better than 5 %, based on calibration to standard solutions and checked against internal laboratory sediment standards. All elemental ratios are normalized to Al, except for organic carbon over total phosphorus (Corg/Ptot), which is reported in mol mol$^{-1}$.

**3.5 Carbonate percentage measurements**

The carbonate concentration of Site 959 samples was determined at the University of Southampton using a UIC CM5015 coulometer operated with a UIC CM5011 emulator and coupled to an AutoMateFX autosampler and carbonate digestion system. The samples were oven-dried at 50°C, crushed with an agate mortar and pestle, placed in a glass vial, and oven dried again after crushing. Each vial was capped immediately after removal from the oven to prevent the samples from taking on moisture. Depending on estimated carbonate content, 10–40μg of bulk dry sample was weighed out on a Sartorius ME5 microbalance and placed in a septum-capped autosampler vial prior to each run. A total of 59 samples were analysed across six runs on separate days, with repeat analyses performed on ~25% of the samples. Each run was calibrated using blank-corrected counts for a calcium carbonate standard (Acros Organics, 99.999% pure) spanning a mass range of 2–10μg. External precision is estimated at ±0.6% $CaCO_3$ based on results from a consistency standard that was included in all runs (average=73.1%; n=12). The detection limit was determined from a compilation of blank analyses from all runs (n=24), and using the average blank counts plus 3×s.d., the detection limit for a typical 15mg sample was calculated to be 0.08% $CaCO_3$.

## 4. Results

### 4.1 Palynology

Dinocysts typically dominate the palynological residues. Pollen and spores derived from terrestrial higher plants are present but in very low abundances (average ~1%; 40 $g^{-1}$ dry sediment). The body of the PETM CIE and the corresponding interval with higher TOC is marked by high abundances of organic linings of benthic foraminifera (Figure 3d). Upper Paleocene dinocyst assemblages are composed of three major components (Figure 2c). The most abundant component is the generalist group *Spiniferites*, followed by *Apectodinium* and Goniodomideae with an epicystal archeopyle (hereafter referred to as Goniodomideae; see Sluijs and Brinkhuis (2009) for taxonomic descriptions of the groups and complexes). Within the uppermost Paleocene, representatives of the *Areoligera* cpx increase in abundance. The onset of the PETM is marked by an acme of *Apectodinium*, reaching highest abundances (>90%) between 803.89 and 803.77 mbsf. The body of the CIE yields extremely low abundances of dinocysts (100-200 gram $^{-1}$) so that percentages are based on counts below 200 specimens and should hence be interpreted as rough estimates (open circles in Figure 2). Regardless, assemblages within the body of the CIE are mostly composed of Goniodomideae, *Spiniferites* and *Apectodinium*. *Senegalinium* and Protoperidinioid cysts occur almost exclusively within the body of the CIE. High cyst concentrations (>$10^4$ gram $^{-1}$) and a series of ~10 subsequent acmes of different dinocyst groups mark the recovery of the CIE (Figure 2c). Towards the top of the analyzed interval, *Florentinia reichartii* becomes the dominant species. Several groups show abundance peaks in the interval following the recovery of the CIE.

### 4.2 Carbonate weight %

Apart from sediments close to the PETM, the studied interval typically comprises 10-30 weight % (wt%) $CaCO_3$ (Figure 3f). However, $CaCO_3$ is below the detection limit in an interval starting just below the CIE, at 804.2 mbsf. Various methods, i.e. weight loss during decalcification, $CaCO_3$ based on ICP-OES analysis of Ca concentrations and dedicated $CaCO_3$

measurements show similar trends (Figure 3f). We use the weight loss during HCl-treatment as a high-resolution approximation for $CaCO_3$ outside the PETM. Especially across the PETM the weight loss during decalcification is strongly influenced by dissolution of HCl-soluble salts and minerals, especially gypsum, which accounts for 5-10% of sediment weight in some parts of the CIE.

## 4.3 Bulk sediment chemistry (XRF and ICP-OES)

We use elemental ratios to reconstruct hinterland hydrology and redox sensitive trace elements to reconstruct bottom water oxygenation for the site location. The Ti/Al ratios from XRF core scanning and ICP-OES analysis show similar trends, with a gradual rise in the upper Paleocene and a second sharp increase at the base of the porcellanite that marks the onset of the CIE (804.1 - 803.8 mbsf, Figure 3g). Hereafter, values drop and remain consistently lower than in the upper Paleocene within the body and recovery phases of the CIE. K/Al ratios correlate well with TOC wt%. Concentrations are slightly higher within the CIE than in the upper Paleocene and interrupted by a sharp drop in the porcellanite layer. The CIE is also marked by elevated concentrations of Cr, S, Ni, V, Zn and progressively higher Corg / Ptot ratios (Cr/Al, V/Al and Corg/Ptot; Figure 3i).

## 4.4 Magnetic susceptibility

Magnetic susceptibility is anti-correlated with carbonate percentages in the upper Paleocene. At 804.2 mbsf, before the onset of the CIE, this relation disappears. The porcellanite layer is marked by low magnetic susceptibility and the body of the CIE is marked by stable high values. A maximum is reached during the recovery phase, at 802.97 mbsf, followed by a second maximum at 802.81 mbsf. Bulk magnetic susceptibility measurements closely resemble Fe concentrations derived from both ICP-OES and XRF (Figure 3j) and correlate to TOC wt% in the upper Paleocene and lower part of the CIE.

## 5. Discussion

## 5.1 Background cyclic variability

The latest Paleocene at the core location is characterized by cyclic variations in proxy records, including total organic carbon content (TOC), dinocysts per gram of sediment, magnetic susceptibility and $CaCO_3$ content (Figure 3). Frieling et al. (submitted) attributed the cycles to climatic precession (21 kyrs). Implied latest Paleocene average accumulation rates are ~1.3 cm $kyr^{-1}$, which is consistent with the available nannofossil biostratigraphy. Cycles may be recorded during the PETM in K/Al ratios (Figure 3g), but these are certainly not unambiguous in other proxy-records that show similar cyclic variations in the Paleocene (e.g. TOC wt% and Fe). However, if these K/Al variations also represent precession, it may imply a two-fold increase in accumulation rate. Cyclic variability is notably absent in dinocyst assemblages, $\delta^{13}C_{TOC}$, $TEX_{86}^{H}$ derived temperatures and Ti/Al ratios. This observation is important since we can, although with caution, use these as indicators of environmental change that is not associated with background cyclic variability. Anomalies in these parameters can then be

used to identify changes in e.g. productivity, sea level or hinterland hydrology that may be related to the PETM, but not astronomically forced.

## 5.2 Depositional setting, sea level, bottom-water oxygen levels

### 5.2.1 Latest Paleocene Background State

Dinocyst assemblages in the Paleocene are characterized by abundant *Spiniferites* spp. and *Apectodinium* spp. Absolute numbers of dinocysts correlate well with TOC wt%, but also with K/Al ratio. In tropical regions, K/Al ratios vary with wet-dry cycles due to variations in sediment provenance, which suggests precession forcing may be an important driver of productivity in the study region. We also find that some of the cycles show a "notch" at the peak, which, given the paleolatitude (3 – 7 ºS), may be interpreted to result from the double overpass of the intertropical convergence zone (ITCZ) (Verschuren et al., 2009).

Occasionally, Goniodomideae are present in great abundance. In the modern ocean (Zonneveld et al., 2013) and in the Paleogene (Sluijs and Brinkhuis, 2009), high abundances of Goniodomideae occur in very shallow marine settings such as lagoons, and are typically associated with warm, stratified waters and very high and/or seasonally fluctuating salinity. In open ocean settings the group can be indicative of strong stratification (Reichart et al., 2004). Considering the offshore location of Site 959, and relatively low abundances of low-salinity-tolerant taxa, we interpret high abundances of Goniodomideae to indicate seasonally strong stratification, either by temperature or salinity.

From 804.4 mbsf, we find an increase in abundance of dinocysts belonging to, or closely related to the genus *Areoligera* (*Areoligera* complex *sensu* Sluijs and Brinkhuis, 2009). A relative abundance increase of this genus was previously interpreted to reflect sea level rise at several shelf and slope sites during the PETM (Sluijs et al., 2008b). However, Site 959 is located in an open ocean setting, which is reflected in the dinocyst assemblages by the very high relative abundance of *Spiniferites*, a genus that is relatively more abundant with increasing distance to coastlines in the Paleogene and the modern (Brinkhuis, 1994; Pross and Brinkhuis, 2005; Zonneveld et al., 2013).

The PETM is associated with a drop in *Areoligera* abundance and a concomitant rise in *Spiniferites* in the relatively offshore locations in New Jersey, such as Bass River (Sluijs and Brinkhuis, 2009), and the Tawanui slope section in New Zealand (Crouch and Brinkhuis, 2005). In the pro-delta settings of ODP Site 1172 on the East Tasman Plateau (Sluijs et al., 2011), the ACEX core in the Arctic Ocean (Sluijs et al., 2008a) and Spitsbergen (Harding et al., 2011), the PETM sees an influx of *Areoligera* into Paleocene dinocyst assemblages dominated by low salinity tolerant *Senegalinium*, evidencing a more marine setting. In the nearby shelf site in Nigeria, *Areoligera* is a common constituent of the assemblage only directly before the PETM (Frieling et al., 2017), perhaps recording both the eustatic rise at the PETM and a latest Paleocene regression (Speijer and Morsi, 2002).

Following inferences from previously published records (Sluijs et al., 2008b), higher sea level should hence result in a decrease in the relative abundance of *Areoligera* at the offshore Site 959. Identifying the cause of this discrepancy is challenging, as the Site 959 record is the first organic-walled dinocyst PETM record from the open ocean. We do not find indications for increase terrestrial input, such as pollen or branched GDGTs, which could indicate the signal was transported from the shelf further offshore and this leaves us to assume these specimens are *in situ*.

However, Goniodomideae, typically associated with stratification (Sluijs et al., 2005; Zonneveld et al., 2013), are abundant lower in the analyzed section and increase in relative abundance together with *Areoligera*. The abundance of Goniodomideae at more offshore localities is commonly interpreted to result from intense stratification (Reichart et al., 2004) and we propose that abundances of *Areoligera* may be explained in a similar way. This inference of strong stratification in the region is supported by data from foraminifera and climate model runs, which indicate the presence of a strong but shallow thermocline in the eastern equatorial Atlantic (Frieling et al., 2017). We therefore speculate that the higher percentages of *Areoligera* here may be related to strong(er) stratification in the latest Paleocene, rather than sea level change. The waters of above the thermocline may have emulated the high-energy environment *Areoligera* prefers (Brinkhuis, 1994; Sluijs and Brinkhuis, 2009).

### 5.2.2 The onset of the PETM and the *Apectodinium* acme

The quasi-global acme of the tropical dinocyst genus *Apectodinium* has long been used as a Paleocene-Eocene boundary marker in biostratigraphic studies (Heilmann-Clausen, 1985; Bujak and Mudge, 1994) and later discovered to be connected to the PETM (Bujak and Brinkhuis, 1998; Crouch et al., 2001). However, some sections show relatively low percentages of *Apectodinium* during the PETM (Sluijs et al., 2006; Frieling et al., 2014).

In addition to the acme of *Apectodinium*, through the entire North Atlantic, Arctic and Northern Tethys, the marker species *Apectodinium augustum* is present during the PETM (Iakovleva et al., 2001; Schmitz et al., 2004; Sluijs et al., 2006; Sluijs and Brinkhuis, 2009; Harding et al., 2011) although other species are often the dominant representatives of the genus. Recent taxonomic revision of several taxa has removed *Apectodinium augustum* from *Apectodinium* and moved it to *Axiodinium* (Williams et al., 2015). However, we follow the comment of Bijl et al. (2016) and retain *A. augustum* within the genus of *Apectodinium*. We consider species within *Apectodinium* to have similar affinities and continue to refer to the quasi-global abundance event as "*Apectodinium*" acme. *A. augustum* was not recorded at Site 959.

The abundance of *Apectodinium* appears to be largely controlled by temperature in higher latitude regions across the PETM (Crouch et al., 2001; Sluijs et al., 2006; Frieling et al., 2014), but it was abundant to dominant in tropical and sub-tropical material already in the Late Paleocene (Jan du Chêne and Adediran, 1984; Crouch et al., 2003; Sluijs et al., 2014; Frieling et al., 2017) and accounts for ~15% of the assemblage at Site 959 above 822.14 mbsf (Awad and Oboh-Ikuenobe, 2016) (Figure 2). Furthermore, the required minimum temperature of ~20 ºC (Frieling et al., 2014) was widespread in many mid latitude sections well before the PETM (Dunkley Jones et al., 2013; Frieling et al., 2017), signaling other environmental

factors were more important in controlling the distribution of *Apectodinium* at low and mid latitudes (Sluijs et al., 2007a; Sluijs and Brinkhuis, 2009).

At Site 959, *Apectodinium* is abundantly present in the latest Paleocene, which is not surprising given the equatorial location. This is similar to the Paleocene in Nigeria (Jan du Chêne and Adediran, 1984; Frieling et al., 2017), Cameroon (Mbesse, 5   2013) and Tunisia (Crouch et al., 2003), indicating *Apectodinium* was already common along African margins before the PETM. More surprising is perhaps the general lack of an *Apectodinium* acme at these low latitude sites during the PETM, although some caution must be placed due to poor preservation at the Tunisian site (Crouch et al., 2003) and the low sampling resolution of the section in Nigeria (Frieling et al., 2017).

At Site 959, we find the highest abundance (95%) of *Apectodinium*, close to the onset of the body of the PETM (803.89 10   mbsf). This is perhaps later than at many mid- and high-latitude sites where the highest abundance of *Apectodinium* often slightly precedes the CIE (Sluijs et al., 2007a, 2011; Kender et al., 2012). However, at many of these sites the assemblages are nearly monospecific and we record the same at Site 959. We thus infer this high abundance of *Apectodinium* to represent the quasi-global *Apectodinium* acme. The high percentages of *Apectodinium* in the porcellanite (804.1 - 803.9mbsf) are most likely mixed in from above (803.85 - 803.75 mbsf), similar to the rest of the organic matter (Frieling et al., submitted), since 15   assemblages and species are very similar.

### 5.2.3 Deoxygenation during the PETM

Many sites globally show decreased bottom-water oxygen content during the PETM and at some shelf sections bottom waters (Dickson et al., 2012, 2014) and even the photic zone became euxinic (Sluijs et al., 2006, 2014, Frieling et al., 2014, 2017; Schoon et al., 2015). Oxygen minimum zones were expanded (Zhou et al., 2016) and deep ocean waters were affected 20   by deoxygenation although anoxia did not develop in the deep sea (Chun et al., 2010; Pälike et al., 2014). At Site 959, we find strong indications of decreased oxygen concentrations in bottom waters in the form of increased organic matter burial fluxes, as assessed through reconstructed accumulation rates and TOC wt%. Moreover, increasing Corg/Ptot ratios (Fig. 3i) most likely relate to preferential regeneration of phosphorus from sediments under anoxic conditions (Slomp et al., 2002; Algeo and Ingall, 2007). The PETM interval is also associated with slightly elevated concentrations of redox sensitive 25   elements (e.g., Cr) (Figure 3i). In apparent contrast is an increase in the abundance of organic linings of benthic foraminifera. Some benthic foraminifera however tolerate low oxygen concentrations for substantial periods (Langlet et al., 2014) and may outcompete metazoans in such conditions (Woulds et al., 2007). This signal is surprisingly similar to that found in shelf sections of the Gulf of Mexico (Sluijs et al., 2014) and Nigeria (Frieling et al., 2017). The combined information from Site 959 and Nigeria suggests that oxygen minimum zones during the PETM expanded upwards onto the shelf and downwards to 30   the paleodepth of Site 959 (>1000 m) in the eastern tropical Atlantic, a phenomenon very similar to modern trends (e.g., (Stramma et al., 2008; Schmidtko et al., 2017).

### 5.2.4 Dinocyst decline during peak PETM

The dinocyst assemblages, but also other microfossil groups, in mid-latitudes have been successfully used for detailed environmental reconstructions (e.g., Gibbs et al., 2006; Hollis, 2007; Sluijs and Brinkhuis, 2009; Stassen et al., 2012). Here, extremely low numbers of dinocysts (100-200 g$^{-1}$, relative to latest Paleocene average of >5000 g$^{-1}$), carbonate and siliceous microfossils during the body of the CIE, which, since accumulation rates increase by at most a factor 2 (Fig. 3g), is appreciably beyond the effect of sediment dilution. Regardless of the cause of eukaryote demise (section 5.4), it hampers a detailed paleoenvironmental reconstruction.

A few dinocyst genera however are present. The CIE is notably marked by higher relative abundances of Protoperidinioid cysts, as well as *Senegalinium* and related genera. Importantly, these are the only groups that increase in relative abundance with respect to the latest Paleocene. Protoperidinioid dinocysts are produced by heterotrophic dinoflagellates in the modern ocean (Jacobson and Anderson, 1986) and most likely also in the Paleogene. Also *Senegalinium* cysts were likely produced by heterotrophic dinoflagellates (Sluijs et al., 2005, 2007a), which implies that a larger proportion of the dinoflagellate production was heterotrophic. This suggests that autrotrophic dinoflagellate species were stressed, but that a food source, locally produced or imported from elsewhere, was abundantly present. Notably, based on empirical information, the species in these groups were tolerant to relatively low salinity (Sluijs et al., 2006; Sluijs and Brinkhuis, 2009; Barke et al., 2011). Although lower Ti/Al ratios may indicate hydrological changes in the hinterlands during the body of the CIE (Figure 3), a large drop in salinity ~150 km offshore, similar to that seen regionally at proximal sites (e.g., Sluijs and Brinkhuis, 2009; Harding et al, 2011), seems unlikely. Low-salinity tolerant dinocysts species, i.e. *Senegalinium* cpx, are abundant in many high latitude sections during the PETM (Sluijs et al., 2006; Harding et al., 2011), but the high relative abundance of Protoperidinioid cysts is only found in two sites in the eastern equatorial Atlantic (Frieling et al., 2017). This is surprising given the high abundances of Protoperidinioid cysts in sediments deposited in nutrient-rich sectors of the modern ocean. Although other groups are consistently present within the peak of the CIE, absolute numbers are much lower than in the latest Paleocene and we cannot exclude that these specimens are reworked or transported over large distances.

### 5.2.5 Recovery of the PETM

At 803.0 mbsf, a small (~1‰) positive step in $\delta^{13}C$ possibly indicates a period of non-deposition or erosion, which is followed by the start of the recovery of the CIE. About ~10cm above the start of the recovery (802.9 mbsf) carbonate accumulation resumes. Dinocyst diversity and abundance however recover synchronous with the $\delta^{13}C$ rise, and even exceed upper Paleocene values. Remarkably, ~10 distinct acmes of different dinocyst groups occur within ~30 cm of sediment within the ~100 kyr recovery phase. These events are hence limited to a maximum duration of 10 kyr, but are likely much shorter since bioturbation smoothed the signal. Moreover, the recovery of the PETM at Site 959 is truncated by a core gap and perhaps preceded by a small hiatus, which likely leads to underestimated accumulation rates in our approach. SST drops concomitantly with the increase in $\delta^{13}C$ (Fig. 3), but does not show a clear relation with any of the individual dinocyst groups. While substantial variability is recorded in high-resolution records of other microfossil groups elsewhere (Giusberti et al., 2016), the combination of extremely short-term and high-amplitude extreme biotic variation is unprecedented for any

PETM site across the globe, certainly for the recovery interval, and indicates highly variable environmental conditions. We observe high transient abundances of cosmopolitan (*Cordosphaeridium*) and generalist genera (*Spiniferites*), species indicative of stratified waters (Goniodomideae), high-energy environments (*Areoligera* cpx) and *Apectodinium* (Figure 2). Although environmental preferences of many of these groups are known, it is difficult and perhaps pointless to disentangle the highly dynamic assemblages into individual environmental signals.

Only the variability of dinocyst assemblages during the 'body' of the CIE at the New Jersey Shelf sites Bass River and Wilson Lake resemble the results at Site 959 (Sluijs and Brinkhuis, 2009). However, Bass River and Wilson Lake are shallower and closer to the paleoshore and therefore more susceptible to environmental swings than the offshore Site 959. The strong variation at Site 959, representing a fully open ocean setting is therefore much more surprising.

Potentially, the recorded variability at Site 959 is related to sub-Milankovitch climate variability, which has been observed in a range of tropical sections from the Proterozoic onwards (Wu et al., 2012; Wilson et al., 2014) and also within the PETM (Abdul Aziz et al., 2008) and in dinocyst records (Reichart and Brinkhuis, 2003). However, we observe no cyclic recurrence of these dinocyst assemblage variations, nor do we find the far more obvious cyclic (precession-scale) variation in the upper Paleocene in the dinocyst assemblages. Therefore this explanation appears unlikely. Unfortunately, none of our proxy records show similar variability in this period so we cannot resolve the underlying cause of such extreme variations in dinocyst assemblages.

### 5.2.6 Post PETM assemblages

Early Eocene dinocyst assemblages and thus presumably environmental circumstances are similar to those found in the Paleocene. Only *Florentinia reichartii* increases towards the top of the analyzed interval. Although this species seems to be associated with somewhat higher temperatures and fresh-water input along the New Jersey shelf (Sluijs and Brinkhuis, 2009), and relates to occurrences of *Apectodinium* at southwest Pacific ODP Site 1172 (Sluijs et al., 2011), it here thrives under apparently relatively stable, open ocean conditions that prevail in the earliest Eocene. In addition to *F. reichartii*, *Apectodinium* is occasionally abundant, as are *Spiniferites* and Goniodomideae. Carbonate content is similar or slightly higher than in the upper Paleocene (10-30%) and bottom waters appear to be well ventilated. Remarkably, the clear correlations between TOC and dinocyst numbers that exists in the upper Paleocene did not return, but dinocyst numbers and K/Al ratios continue to show cyclicity at similar frequency to that in the upper Paleocene, suggesting the same forcing, possibly precession driven hydrological changes, remained an important factor governing plankton assemblages at Site 959 (see also section 5.2.1).

### 5.3 Absolute temperature and temperature change in a global perspective

The recorded warming across the onset of the PETM (3.9 ºC) is similar to the global average (4-5 ºC, e.g., Dunkley Jones et al., (2013); Frieling et al., (2017)). However, the few sampled tropical regions typically warmed slightly less (Tripati, 2003; Zachos et al., 2003; Kozdon et al., 2011) than global average due to persistent extratropical amplification of temperature

change (Frieling et al., 2017). The isoprenoid glycerol dialkyl glycerol tetraether (iGDGT) distribution we find here is somewhat similar to modern Red Sea distributions. Within the modern Red Sea, $TEX_{86}$ behaves slightly differently: the relation with temperature has a different slope and is offset from the global calibration (Trommer et al., 2009). The linear Red Sea calibration of Trommer et al., (2009) yields a warming of 3.4 ºC, only slightly smaller than the 3.9 ºC recorded by

$TEX_{86}^{H}$, which uses a global calibration dataset (Kim et al., 2010). Furthermore, although iGDGT distributions may look similar, there is no reason to assume that Site 959 was subject to a similar setting or that Thaumarcheota communities were similar to those in the modern Red Sea. We note that Paleocene $TEX_{86}^{H}$ reconstructed absolute temperatures are in the same range as those previously reported from this region (Frieling et al., 2017), which are supported by planktonic foraminifer $\delta^{18}O$ and Mg/Ca derived temperature estimates from the same section. Even though these temperatures are outside the

calibration interval (5-30 ºC) for $TEX_{86}^{H}$, the relation with temperature continues to at least 40 ºC, based on mesocosm culture experiments (Wuchter et al., 2004; Schouten et al., 2007). Importantly, the maximum temperature that can be reconstructed by $TEX_{86}^{H}$ is 38.6 ºC. The reconstructed temperatures are close to this maximum and we may actually approach the limit of the proxy and underestimate maximum PETM surface-water temperatures (Frieling et al., 2017; submitted).

**5.4 Heat-stress and the demise of eukaryotes**

**5.4.1 Site 959 and Nigerian Shelf**

Sediments typically contain less than 30% $CaCO_3$ in the latest Paleocene, but already from 804.2 mbsf, just below the onset of the CIE, $CaCO_3$ is completely absent (Figure 3f). We explore several factors that might explain this feature. First, similar to Nigeria (Frieling et al. 2017), water column deoxygenation at Site 959 is asynchronous with the demise of dinoflagellates

and other microfossil groups and must therefore be decoupled. Second, it could be due to post-depositional carbonate dissolution through PETM ocean acidification as recorded in deep ocean basins (Zachos et al., 2005). However, at no single site described so far, the carbonate compensation depth (CCD) rose above the reconstructed paleodepth (~1000m, Oboh-Ikuenobe et al., 1997) for this site, so this scenario seems unlikely. Third, carbonates can be dissolved locally through enhanced oxic organic matter (OM) decomposition at the sea floor, producing $CO_2$. However, the dinocyst concentrations

(~$10^4$ gram$^{-1}$) or TOC (~0.5%) content in this interval are not anomalous compared to the entire upper Paleocene, so this factor is unlikely the sole explanation. Finally, there may be a general suppression of carbonate (export-) production because of biotic stress, either through acidification or warming (Aze et al., 2014; Frieling et al., 2017). We surmise that low carbonate production, possibly associated with the (pre-CIE) warming ((Frieling et al., submitted), Figure 3c) played a pronounced role in the absence of carbonate throughout the PETM.

After carbonate accumulation ceased, without apparent change in other proxies or large changes in sediment accumulation rates, dinocyst concentrations decrease from 5000 g$^{-1}$ to 100-200 g$^{-1}$ at 803.70 mbsf (Figure 3d). Organic linings of benthic foraminifera continue to increase in abundance indicating that export productivity did not collapse. This signal is identical to that found in a shelf section from Nigeria (Frieling et al., 2017); both sections show very low (100 - 200 g$^{-1}$) abundances of

dinocysts during peak PETM warmth. We also note that pyritized biogenic silica, which is abundantly present in the upper Paleocene and early Eocene, is largely absent above the porcellanite and only reappears during the recovery (Figure 3e).

In addition to these overall similarities, the dinocyst assemblages at Site 959 are very similar to those found at the PETM in Nigeria. Both sections are marked by relatively high percentages of Protoperidinioid dinocysts within the body of the PETM,

which is an exclusively heterotrophic group (Jacobson and Anderson, 1986). Oxidation experiments show Protoperidinioid cysts are usually less resistant to oxidation than, e.g. *Spiniferites* cysts (Zonneveld et al., 1997, 2008). However, the absence of a relation between the abundance of these taxa and TOC content suggests that preservation is not a major factor. We speculate that these species resided in slightly deeper waters, possibly near the thermocline.

The combined information suggests that eukaryote activity in the mixed layer was suppressed not only in Nigeria (Frieling et

al., 2017) but also at the more offshore Site 959. Similar to Nigeria, there is hence no evidence that the low numbers of dinocysts resulted from severe stratification and anoxia, strong variations in salinity or biases resulting from preservation. Although the initial absence of carbonate is perhaps related to a secondary factor, we attribute the drop in eukaryote production in this region to heat stress, as this is among the few physicochemical factors that can generate the same effect in both the open ocean (Site 959) and on the shelf (Nigeria).

It is important to note that primary productivity did not collapse due to the lack of eukaryotes. Organic linings of benthic foraminifera are abundant within the PETM interval at Site 959 and in Nigeria, which suggests that food supply was likely similar as before the PETM. Both sections are characterized by enhanced organic carbon content and burial. This apparently prokaryote dominated ecosystem, with enhanced organic carbon burial hence bears some similarities with tropical ecosystems during the Cretaceous Ocean Anoxic Events (OAEs) (Kuypers et al., 2001).

**5.4.2 Global tropical heat stress?**

While heat-stress seems the most likely option for the demise of eukaryotes in the equatorial Atlantic, other open ocean equatorial and tropical (20 ºS – 20 ºN) sites (Table 1) apparently do not show such effects, although some of these records are affected by dissolution and most are devoid of organic carbon. In the tropical and equatorial Pacific (ODP Sites 865, 1209, 1220 and 1221) no obvious decrease in calcifying planktic eukaryotes is found during the PETM, but we note that SST

reconstructions from these sites (Tripati, 2003; Zachos et al., 2003; Kozdon et al., 2011) do not show the excessively high temperatures (>35ºC) recorded at Site 959, Nigeria or Tanzania (Aze et al., 2014; Frieling et al., 2017). Lower calcium carbonate accumulation at these sites has been explained by dissolution. Also the absence of calcium carbonate at sites in the Caribbean and western equatorial Atlantic (ODP Sites 999, 1001, 1258 and 1260) has been attributed to carbonate dissolution (Bralower et al., 1997; Mutterlose et al., 2007). Importantly, however, an influence of heat-stress on carbonate

production cannot be excluded. In contrast to these relatively deep open ocean sites, the southern Tethyan shelf and slope (Egypt) certainly was not affected by CCD rise. However, the PETM in Egypt is often marked by relatively organic rich shales mostly devoid of carbonates (Schulte et al., 2011) and yet, as Speijer and Wagner (2002) note, contain no dinocysts, spores or pollen. These observations could be interpreted as supportive evidence for heat-stress among planktic eukaryotes

and possibly also land-plants (Huber, 2008). Aze et al., (2014) hypothesized heat-stress effects may have played a role in Tanzania during the PETM as coccolithophores and foraminifera both decline in abundance and although calcareous dinoflagellate cysts increase in relative abundance, also these remain at very low absolute abundances (Bown and Pearson, 2009). We surmise that heat-stress effects during the PETM may not have been limited to the eastern equatorial Atlantic and

western Indian Ocean, although it is difficult with the presently available data to disentangle the effects of dissolution, temperature and other environmental stressors at all potentially affected sites.

## 5.5 Potential influence of heat-stress on ocean carbonate chemistry

It is far from certain if heat-stress affected planktic calcifiers not only in the eastern equatorial Atlantic and the western Indian Ocean, but in larger areas in the tropics. However, if it did, it might have affected sedimentary sequences in the deep

sea to the extent that it had implications for global carbon cycling during the PETM and thus calculations of the mass of carbon that was injected into the ocean-atmosphere system during the PETM (e.g., Panchuk et al., 2008; Zeebe et al., 2009; Luo et al., 2016).

First, it would imply that the reduction in carbonate accumulation rates at tropical sites were potentially a combined result of acidification and heat-stress. If so, the reduction in carbonate accumulation across the PETM should have been relatively

high at tropical sites relative to higher latitudes (see section 5.4). Interestingly, the reduction in carbonate accumulation was most severe at Walvis Ridge (Zachos et al., 2005), and the Caribbean (Bralower et al., 1997), both located towards the borders of the tropical band, which is in sharp contrast to sustained carbonate deposition in the Atlantic Southern Ocean sites 689 and 690 (Kelly et al., 2005, 2010). Although carbonate wt% at Pacific tropical sites remain high (Colosimo et al., 2006; Leon-Rodriguez and Dickens, 2010), this is largely due to very low accumulation rates of the sole other sedimentary

component, clay, and bioturbation, complicating robust estimates of carbonate accumulation through time.

Second, because calcification consumes alkalinity and produces $CO_2$, heat-stress limited carbonate production would ultimately act as a negative feedback to acidification following the PETM carbon injection. On a global level, it would ultimately lead to accumulation of alkalinity in ocean water. As such, it could be seen as an indirect, although potentially important component of so-called biological carbonate compensation: a reduction in carbonate production due to

acidification (Luo et al., 2016). This has been invoked as an alternative model to explain the CCD overshoot, i.e., the fact that $CaCO_3$ wt% within the recovery phase of the PETM exceeds that of the late Paleocene in many deep sea sections(e.g. Luo et al., 2016; Penman et al., 2016).

Collectively, it requires thorough reinterpretation of numerous published records to evaluate to what extent tropical heat stress might have contributed to reduced carbonate accumulation in the global ocean.


## 6. Conclusions

The Paleocene – Eocene transition at Site 959 is marked by a warming of 3.9 ºC. Absolute $TEX_{86}^{H}$ temperatures at 37.4 ºC are unprecedented for the PETM, although similar to recent estimates from Nigeria and in good agreement with climate model estimates.

Importantly, based on our multi-proxy records, we conclude that these extreme temperatures caused a remarkable drop in dinocyst abundances, and most likely also in the production of biogenic carbonate and opal during the PETM. Crucially, only heterotrophic dinoflagellate species, which likely resided somewhat deeper in the water column, persisted. Together with relatively abundant organic benthic foraminifer linings this indicates sustained primary production, likely dominated by prokaryotes. If the drop in calcification was global, it might have contributed to the recorded decline in deep sea carbonate

accumulation, and, as such, it is an important factor to constrain because it potentially affects calculations of global ocean acidification and $CO_2$ input.

Combined evidence from dinocysts and inorganic chemistry points to an altered hydrological cycle at the PETM, resulting in fluctuations in stratification. Furthermore, progressively less oxygenated bottom waters characterize the body phase of the PETM. In sharp contrast to the general absence of eukaryotes during the body phase of the PETM, the recovery phase of the

PETM is highly dynamic, with several groups of dinocysts dominating assemblages on timescales shorter than 10 kyr.

*Author contributions.*

J.F. and A.S. designed the study. J.F., S.M.B., J.J.M., U.R., T.W. and G.-J.R. performed inorganic geochemical analyses. J.F. generated palynological data. All authors analyzed and discussed the data. J.F. and A.S. wrote the paper, with input from all authors.

*Data availability*

Data used in this publication will be made available through online database Pangaea (https://www.pangaea.de/)

*Acknowledgements.*

We thank N. Welters, A. van Dijk, T. Zalm and D. Kasjaniuk (Utrecht University) for analytical support and A. Wülbers and W. Hale (IODP Bremen Core Repository) and W. Schinkel (Erasmus University Rotterdam) for sampling assistance. The European Research Council (ERC) under the European Union Seventh Framework Program provided funding for this work by ERC Starting Grant 259627 to A. Sluijs. The Netherlands Organization for Scientific Research (NWO) supported this work through grant #834.11.006 to G.-J. Reichart. The Deutsche Forschungsgemeinschaft supported U. Röhl and T. Westerhold. This work was carried out under the program of the Netherlands Earth System Science Centre (NESSC), financially supported by the Ministry of Education, Culture and Science (OCW). The International Ocean Discovery Program (IODP) is acknowledged for access to materials and data.

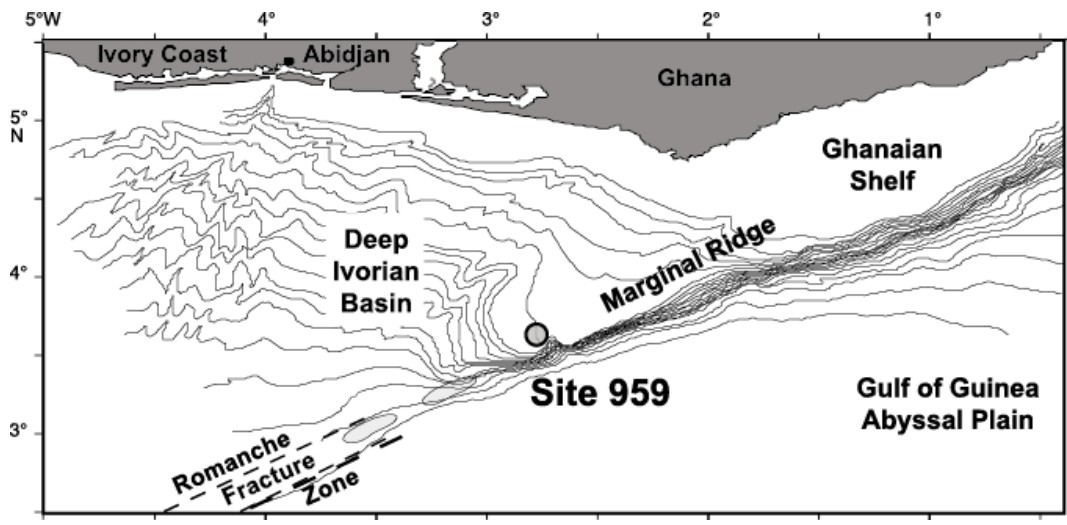

**Figure 1.** Map of Eastern Tropical Atlantic and location of ODP Site 959.

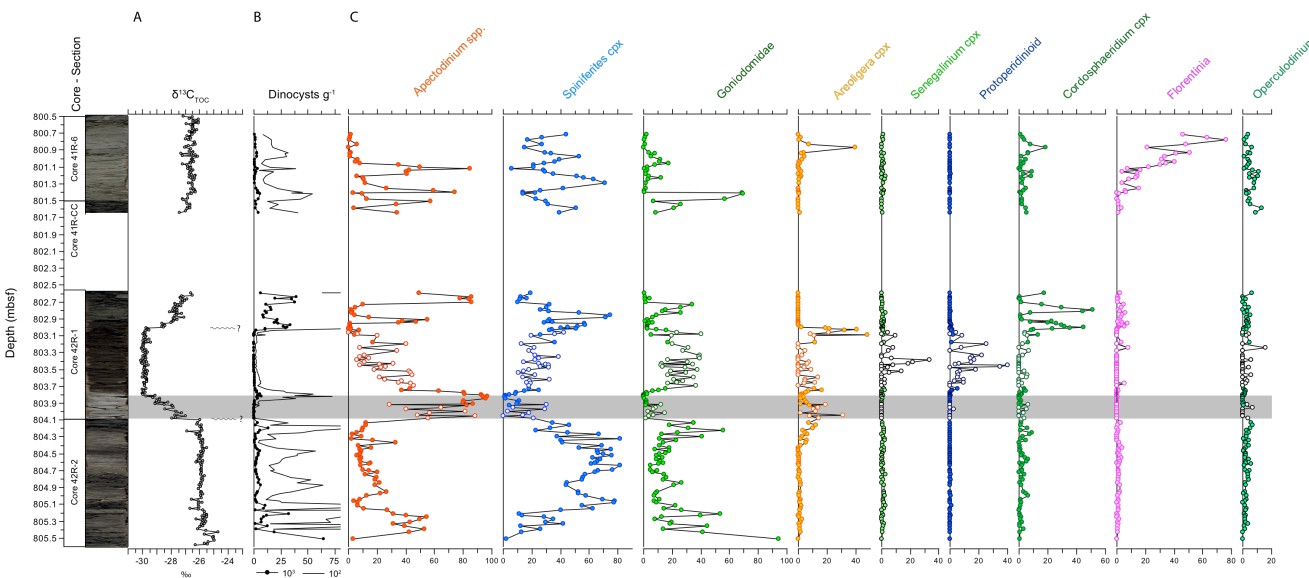

**Figure 2.** The PETM as recorded by dinoflagellate cyst assemblages at Site 959. **A.** $\delta^{13}C_{TOC}$ in permil. **B.** Concentration of dinocysts per gram dry sediment. **C.** Dinocyst groups in percentage of total dinocysts. The grey band marks the porcellanite. Symbols in white indicate counts <100.

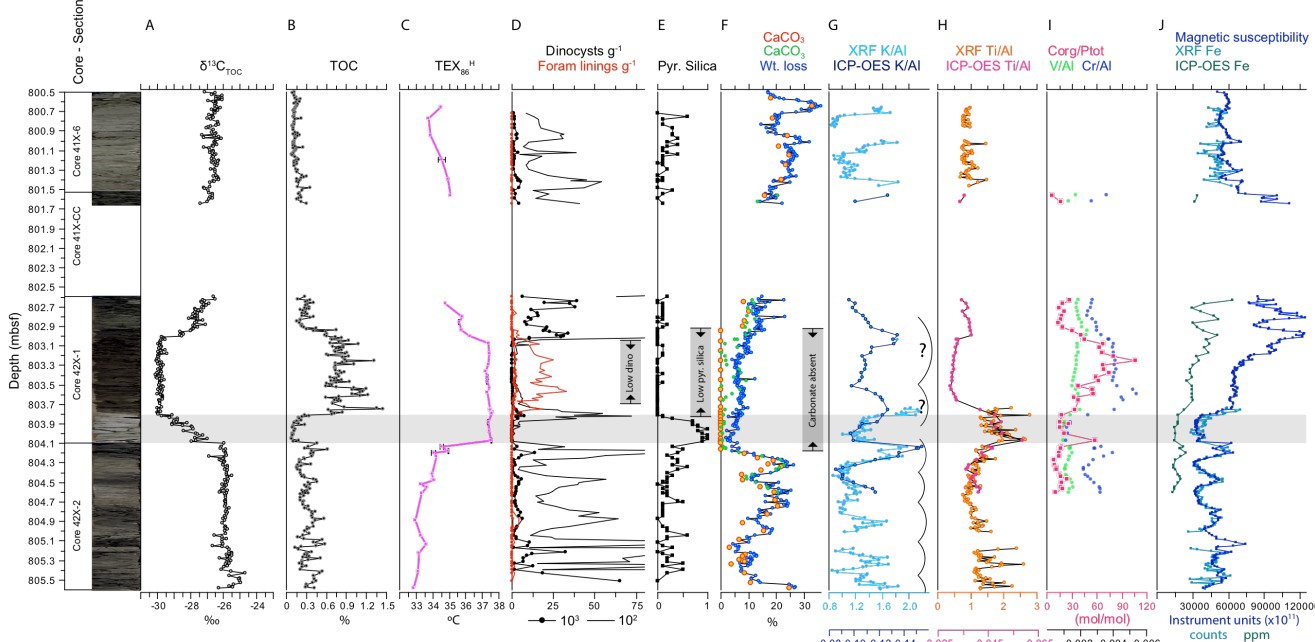

**Figure 3.** The PETM as recorded by organic and inorganic geochemical proxies at Site 959. **A.** $\delta^{13}C_{TOC}$ in permil. **B.** Total organic carbon (TOC) in weight percentages. **C.** $TEX_{86}^{H}$ derived temperature. **D.** Dinoflagellate cyst and organic linings of benthic foraminifera in n g$^{-1}$ of dry sediment. **E.** Fraction of pyritized silica particles in palynological residue. **F.** Weight loss during decalcification (wt. loss) as indication of $CaCO_3$ content; $CaCO_3$ calculated from Ca concentration ICP-OES (green) and selected samples analyzed for $CaCO_3$ on a coulometer (orange). **G.** K/Al ratio from XRF (light blue) and ICP (dark blue). **H.** Ti/Al ratio from XRF (orange) and ICPOES (pink). **I.** Cr/Al (blue) and V/Al (light green) and Corg/Ptot ratio (pink) in mol mol$^{-1}$. **J.** Magnetic susceptibility in instrument units (x10$^{11}$) (dark blue), XRF Fe counts (light blue) and ICP-OES Fe concentrations in ppm (green). The gray band (804.1 – 803. 83 mbsf) marks the porcellanite. $\delta^{13}C_{TOC}$, TOC and $TEX_{86}^{H}$ data from Frieling et al., submitted.

| Site/Section name (references) | PETM SST (°C) | SST Proxy | Oceanic Basin | Paleo-depth (m) | Evidence for heat-stress? |
|---|---|---|---|---|---|
| ODP 959 (this study) | 37.4 | $TEX_{86}^{H}$ | Atlantic | ~1000 | yes, decrease in dinoflagellates, carbonate & biosilica |
| Dahomey Basin, Nigeria (a) | 36.1 | $TEX_{86}^{H}$ | Atlantic | shelf | yes, decrease in dinoflagellates and mixed-layer foraminifera |
| Tanzania TDP14 (b,c) | >35.4 | $\delta^{18}O$ | Indian | shelf | yes, decrease in coccolithophores and mixed-layer foraminifera |
| ODP 1209 (d) | 33.5 | Mg/Ca | Pacific | ~2000 | no |
| ODP 865 (e,f,g, h) | 33.2 | Mg/Ca $\delta^{18}O$ | Pacific | 1100-1300 | no |
| ODP 999 & 1001 (i) | - | | Atlantic | 1500-2500 | combination of dissolution and heat stress? |
| ODP 1215, 1220, 1221 (j) | - | | Pacific | ~2500 | no |
| ODP 1260 (k) | - | | Atlantic | ~2500 | combination of dissolution and heat stress? |
| Multiple sites in Egypt (l) | - | | Tethys | shelf/slope | ambiguous: mixed-layer foraminifera present, but no dinoflagellates, diatoms or pollen |

**Table 1.** Compilation of PETM sites that are tropical or show evidence for heat-stress. References: a (Frieling et al., 2017), b,c (Bown and Pearson, 2009; Aze et al., 2014), d (Zachos et al., 2003), e,f,g,h (Kelly et al., 1996, 1998; Kozdon et al., 2011; Yamaguchi and Norris, 2015), i (Bralower et al., 1997), j. (Raffi et al., 2005), k. (Mutterlose et al., 2007), l. (Speijer and Wagner, 2002). Abbreviations: Sea surface temperature (SST), Ocean drilling program (ODP), Tanzania drilling program (TDP).

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
