# Peer review of "Tropical Atlantic Climate and Ecosystem Regime Shifts during the Paleocene-Eocene Thermal Maximum"

_Climate of the Past, 2017_

## Referee Comment (RC1) · Anonymous Referee #1 · 25 Jul 2017

Dear Editor,

I have now read the manuscript by Frieling et al. entitled "Tropical Atlantic Climate and Ecosystem Regime Shifts during the Paleocene-Eocene Thermal Maximum". The manuscript presents new dinocyst and sedimentological data from the Paleocene/Eocene thermal maximum as recorded at tropical Site 959. The present paper is companion of another paper on the same record submitted somewhere else, and in its main conclusion follows up what previously found by Frieling et al. (2017) at a tropical shallow marine section deposited in Nigeria. In this paper the authors suggest that extreme warming during the PETM within this shallow and possibly restricted basin knocked off planktonic eukaryotic productivity. In the manuscript under consideration they came to the same conclusion, arguing for a collapse of marine eukaryotic productivity due to extreme temperatures also at the fully open ocean setting of Site 959. This represents the main finding of the paper. This is an interesting finding as it highlights how in the open ocean temperature alone may threats pelagic food webs, especially at low latitudes. On the other hand, the paper in my opinion suffers from the fact that several important aspects concerning the record of the PETM at the studied site are shown but not treated in this manuscript. The reader throughout the text is referred to the companion paper that is still – at least to my knowledge – under revision/submitted somewhere else. Since the PETM from Site 959 has never been described before, this in my opinion hampers a full overview of the environmental/oceanic conditions at Site 959 and especially on how the PETM was expressed at this locality, and somehow weaken the author main findings. The sedimentary and isotopic records of the PETM at Site 959 are quite peculiar, and to my knowledge are not alike anyone else found. However what this might mean in term of depositional environment, water column conditions, and in relation with the biotic data presented is insufficiently discussed here, and the reader is referred to the companion paper.

I think this paper could be published after some substantial revisions. However I also believe this paper would acquire more strength and relevance if published once the companion paper will be published, when the authors will be free to openly discuss their other findings.

Line by line remarks:

Abstract:

Lines 23-25: *"The early stages of the PETM are marked by a typical acme of the tropical genus Apectodinium, which reaches abundances of up to 95%. Subsequently, dinocyst abundances diminish greatly, as do carbonate and pyritized silicate microfossils"*
Dinocyst absolute abundance drops already within the porcellanite layer coinciding with the onset of the CIE at Site 959. Why you do not consider it at all? This should be explained and discussed in the text.

Introduction:

Lines 10-13 pag 2: *"The CIE has a distinct shape; a rapid "onset" (1-5 kyr; (Kirtland Turner and Ridgwell, 2016; Zeebe et al., 2016) followed by a prolonged (50-70 kyr) period, the "body", of stable low 13C values and a recovery that lasts 42 - 100 kyr (Röhl et al., 2007; Abdul Aziz et al., 2008; Murphy et al., 2010) to values that remain slightly 13C-depleted (0.5-1 ‰) relative to the latest Paleocene"*.

Not correct. The cited papers estimate a duration of the CIE "body" of 70-100 kyr and of 100-130 kyr for the CIE recovery interval, please amend.

Line 19 pag 2: More reference needed here as the quoted paper is about a tropical locality.

Line 20 pag 2: A widespread expansion of suboxic to anoxic waters during the PETM has been suggested by several works. Some more should be mentioned here, e.g. particularly relevant are: Nicolo et al., 2010; Zhou et al., 2014; 2016; Stassen et al., 2015.

Line 21 pag 2: See also Giusberti et al. 2016.

Line 24 pag 2: Please add some more recent work (a number of paper is likely to have been published on this topic since 2014).

Material

Lines 23-25 pag 3: A more in depth description of how the PETM is identified at Site 959 and how it compares to other records should be part of this paper as well. It does not suffice to refer to the companion paper submitted.

Lines 3-5 pag 4: *"The organic-lean biogenic silica and siliciclastics (804.1 – 803.8 mbsf) are likely derived from an allochthonous, potentially turbiditic, deposit, as reflected by anomalous Ti/Al ratios (Frieling et al., submitted), and obscures the exact onset of the carbon isotope excursion (CIE)"*.

Why the authors argue that this layer has an allochthonous origin? Ti/Al per se does not indicate reworked material. To the opposite a number of evidence would suggest an autochthonous origin (gradually decreasing δ13C values, the drop in dinocyst abundance, a peak in pirityzed microfossils) possibly linked to the peculiar (extremely hot? euxinic?) seafloor and water column conditions at the onset of the PETM. This layer coincides with the onset of the CIE at Site 959, therefore its nature and possible origin and relation with the PETM should be better explained and discussed.

Lines 7-9 pag 4: *"The gradual decrease from ~ -27‰ at 804.09 mbsf to -30‰ at 803.8 mbsf is interpreted as a mixing line between organic matter produced during the Paleocene and the PETM (Frieling et al., submitted)"*.

Mixing of pre-PETM and PETM organic matter would produce a flat line within the porcellanite layer, with isotopic values intermediate between those of the upper Paleocene and the early

Eocene. Infact, the observed continuous decrease in carbon isotope values suggests the original onset of the CIE (or at least part of it) was being recorded. The CIE onset at Site 959 is somewhat similar to the onset of the CIE as recorded at Wilson Lake, New Jersey, see Stassen et al. 2015. This possibility should be explored, and the authors should better support their argument. If the porcellanite layer does record the onset of the CIE, this would imply an autochthonous origin for the layer. Why the authors so sharply disregard the possibility of an autochthonous origin?

Methods

The methods section must include a description of the age model mentioned in the discussion. Changes in accumulation rates affect microfossil absolute numbers, so an age model constraining accumulation rates must be described and be part of this work. Furthermore, this section must include how dinocyst absolute numbers were calculated.
Also, the section lacks a description of the organic geochemistry methods used for lipid extraction and of the calibrations used to convert TEX86 into temperature. Since TEX86 data are discussed, the methods should provide this information as well.

Results

The results section should include a description of TEX86 data and the BIT index must be shown together with TEX86. The BIT index and its meaning in relation with TEX86 should then be discussed in the Discussion paragraph.

Line 17 pag 6: *"The onset of the PETM is marked by an acme of Apectodinium…"*
In the result description the authors should refer to their (local) CIE signal rather than to the PETM, and they should always clearly keep this distinction. There should then be a section in the discussion in which the authors explain how they correlate the CIE at Site 959 to the PETM.

Line 18 pag 6: Not only the body of the CIE yields low abundances of dinocyst, the drop is firstly observed within the porcellanite layer. The author must explain why this is not taken into account.

Line 22 pag 6: There should be "across" instead of "during".

Discussion:

Lines 28-29 pag 7: *"This observation is important since we can, although with caution, use these as indicators of environmental change that is not associated with background cyclic variability".* The authors should further explore this.

Line 21-23 pag 9: *"The high percentages of Apectodinium in the porcellanite (804.1 - 803.9mbsf) are most likely mixed in from above (803.85 - 803.75 mbsf), similar to organic matter (Frieling et al., submitted), since assemblages and species are very similar"*

Again, why it could not be an original signal? Abundance peaks of *Apectodinium* are recorded almost everywhere at the onset of the PETM.

Line 24 pag 9: paragraph title: *"5.2.3 Microfossil decline and deoxygenation during peak PETM"* What's the peak of the PETM at Site 959? The author never states what they interpret to be the peak PETM at the studied site. This should be part of the discussion I mentioned above in this revision.

Lines 26-27 pag 9: Accumulation rates are mentioned but an age model is not described in the methods. It looks like the period misses a verb.

Lines 2-4 pag 11: *"No such short-term extreme biotic variation is known for any PETM site across the globe, certainly not for the recovery interval, and indicates highly variable environmental conditions."*
This is very poorly phrased and can generate confusion. Short term extreme biotic variations are widespread observed, and they represent one of the peculiar features of the PETM as well as one of the reasons why it is so much studied. Think only to the abrupt benthic foraminiferal extinction event at the onset of the PETM. Besides, short term biotic variation across the recovery phase have been observed also at other localities, in particular at shallow/marginal settings, e.g. see Luciani et al., 2007; Stassen et al., 2015; Giusberti et al., 2016.

Lines 10-11 pag 12: *"The combined information suggests that eukaryote activity in the mixed layer was suppressed not only in Nigeria (Frieling et al., 2017) but also at the more offshore Site 959. Similar to Nigeria, there is hence no evidence that the low numbers of dinocysts resulted from severe stratification and anoxia..."*
In a previous paragraph the authors state: *"At Site 959, we find strong indications of decreased oxygen concentrations in bottom waters in the form of increased organic matter burial fluxes, as assessed through reconstructed accumulation rates and TOCwt%. Moreover, increasing Corg/Ptot ratios (Fig. 3i) relate to preferential regeneration of phosphorus from sediments under anoxic conditions (Slomp et al., 2002; Algeo and Ingall, 2007)..... The combined information from Site 959 and Nigeria suggests that oxygen minimum zones during the PETM expanded upwards onto the shelf and downwards to the paleodepth of Site 959 (>1000 m) in the eastern tropical Atlantic, a phenomenon very similar to modern trends (e.g., Stramma et al., 2008). "*.

Please make your data interpretation consistent.

Typing Errors:

Fig. 3: Core sections should be named 42R-2, 42R-1 etc instead of 42X-2 etc?

Line 6 pag 7: "a" missing

Line 1 pag 9: "the" genus *Apectodinium.*

---

## Referee Comment (RC2) · Anonymous Referee #2 · 28 Jul 2017

Reviewer 2 Comment on Frieling et al. 2017. Tropical Atlantic Climate and Ecosystem Regime Shifts during the Paleocene-Eocene Thermal Maximum. Clim. Past Discuss., https://doi.org/10.5194/cp-2017-76

This contribution provides detailed new dinoflagellate assemblage data through the PETM from the western tropical Atlantic, supported by additional new bulk sediment chemistry and magnetic susceptibility measurements. Overall it is clearly presented, well written and a solid contribution to the dataset of surface ecosystem responses to the PETM warming event. I do have a couple of concerns and some minor comments that need to be addressed by the authors before publication.

1. One of the key points made within this paper is that, to quote from the abstract: "The combined paleoenvironmental information from Site 959 and a close by shelf site in Nigeria implies the general absence of eukaryotic surface-dwelling microplankton during peak PETM warmth is most likely caused by heat stress."

My concern is that evidence presented from selected sites is framed to make inferences about global responses and environmental drivers: "Site 959 and a close by shelf site in Nigeria. . ... implies the general absence of. . .". Within a few words they've gone from local and specific ("close by") to a "general absence".

This is a problem because this group of authors are foremost in the analysis of PETM dinoflagellete (and other) records. The quality of their regular outputs, and regard within the community, gives them a very strong influence on shaping the accepted narrative and interpretation of data. With this is mind, I think they have to be exceptionally careful about the claims that are made and that these fully take into account uncertainty in going from the observed data to interpretation.

In this case they may well be correct and are presenting a substantive account of the true ecosystem responses, but my concern is that only references that support this "heat stress" and tropical exclusion of eukaryotes are cited – self-citations and Aze et al. (2014) and Yamaguchi & Norris (2015). A stronger case would include a wider overview and consideration of tropical sites where there is less or no evidence for the exclusion of eukaryotes. For example, the Tanzanian section discussed by Aze et al. (2014) also has records of coccolithophore communities and calcareous dinoflagellates throughout the PETM – calcareous dinoflagellates are actually shown to increase in abundance during the PETM (Bown and Pearson, 2009). Similar records of persistence of coccolithophore communities and increase in calc. dinos are shown from the tropical Pacific, ODP Site 1209 (Gibbs et al., 2006b). In Site ODP 1209 there is an increase in phytoplankton turnover (Gibbs et al., 2006a), which may be related to heat stress, but there is little evidence for a total exclusion of eukaryotic microplankton from this tropical location. There may be reasons for this increase in calc. dinos. in both

the Tanzanian and Pacific tropical sites, and this might support some of this groups' interpretations, but there needs to be some recognition that these other records exist and then an integration of data to form a more solid interpretation of the wider (/global) patterns of change.

In this instance, is there a case for any ecological exclusion of dinoflagellates be limited to the (eastern) equatorial Atlantic? I don't think there is strong evidence (yet) to extrapolate from these two relatively close sites (Nigeria and ODP 959) to a global response in the tropical oceans. Any associated sea surface temperature records from these locations might also just represent localized effects that aren't replicated in either the tropical Pacific or Indian Oceans.

2. The use and referencing of a submitted manuscript "Frieling et al. submitted" is frustrating. This was not provided to reviewers. Although I don't think the conclusions of this manuscript rely on what may be contained within this other submission, one feels that we're being asked to review this paper with 20% of the interpretation (and data?) hidden from view. Ideally, I would rather this manuscript was not published until either the "submitted" manuscript was published or made available for reviewers and editors of this submission. For example, key interpretation of the CIE, its onset and the temperature data are all likely contained in this other submission. I would recommend that the editor at least be able to see this other submitted manuscript in confidence prior to any final publication of this paper, so that they can judge the degree of overlap.

3. Related to the development of a narrative for PETM dinoflagellate records presented by this group over a number of years, I'm intrigued by the interpretation presented of changes in abundance of key indicator species that previously have been used to infer sea level change through the PETM in shelf sites (page 8).

"From 804.4 mbsf, we find an increase in abundance of dinocysts belonging to, or closely related to the genus Areoligera (Areoligera complex sensu Sluijs and Brinkhuis, 2009). A relative abundance increase of this genus was previously interpreted to reflect sea level rise at several shelf and slope sites during the PETM (Sluijs et al., 2008). However, Site 959 is located in an open ocean setting, which means water depth and shore proximity proportionally do not change as much as may be expected from sites on the continental shelf, especially if estimates of the amplitude of sea level rise across the onset of the PETM (5-20m, e.g., (Speijer and Morsi, 2002; Sluijs et al., 2008) are considered. The increase in Areoligera is further associated with a decrease in Spiniferites, consistent with other PETM records (e.g., Sluijs et al., 2008), including a recently published record from Nigeria (Frieling et al., 2017). Since we cannot distinguish between transported and local signals, we may either record a signal that is transported off the shelf, or a local signal that is similar to, but not related to sea level."

I find this a little odd. If the dinoflagellate records are so subject to transport across shelf to the slope and deep ocean, what use are they in reconstructing relative position, from the marginal to oceanic? Which I thought was a substantial component of dinoflagellate paleoenvironmental interpretations? The other option presented is that this assemblage change is: "similar to, but not related to sea level." This seems more likely than pervasive long distance transport. But if there is an alternate environmental cause of this assemblage change in the open ocean sites, then doesn't this also somewhat question whether the interpretation - of the same assemblage changes through the PETM from shelf-records - as being caused by sea-level is open to some reinterpretation? Could there rather be a broader dinoflagellate assemblage change (increase in Areoligera) that is rather related to the wider environmental changes in the tropical / sub-tropical Atlantic and less controlled by sea level? If there are such major PETM environmental changes in the tropics / sub-tropics, such as the heat stress the authors propose plus potential changes in stratification and nutrient supply, wouldn't these be more likely the drivers of dinoflagellate assemblage changes than a relatively modest change in sea level? If so, then this seems like an appropriate place to put the previous interpretations into this new context for the non-expert reader. Again, I'd emphasize, that when this group of authors dominate the generation of PETM dinoflagellate records and the interpretation of them, it's also their responsibility to the external

readership to directly address such questions as new data and interpretations arise.

Other comments:

1. Given that Thomas Westerhold is a co-author, I'm surprised that there is no mention, use or citation of the latest age model assessment for the PETM: Westerhold et al. 2017. Clim. Past Discuss. https://doi.org/10.5194/cp-2017-74. And specifically the durations provided for the PETM in this paper appear to be at odds with Westerhold et al. 2017.

2. Use of capitalization for informal sub-epochs / sub-series: e.g. Page 2, line 3: "during the Late Paleocene and Early Eocene…". See Pearson et al. 2017. Episodes, http://dx.doi.org/10.18814/epiiugs/2017/v40i1/017002

Bown, P., and Pearson, P., 2009, Calcareous plankton evolution and the Paleocene/Eocene thermal maximum event: New evidence from Tanzania: Marine Micropaleontology, v. 71, no. 1-2, p. 60-70.

Gibbs, S., Bown, P., Sessa, J., Bralower, T., and Wilson, P., 2006a, Nannoplankton extinction and origination across the Paleocene-Eocene Thermal Maximum: Science, v. 314, no. 5806, p. 1770.

Gibbs, S., Bralower, T., Bown, P., Zachos, J., and Bybell, L., 2006b, Shelf and open-ocean calcareous phytoplankton assemblages across the Paleocene-Eocene Thermal Maximum: Implications for global productivity gradients: Geology, v. 34, no. 4, p. 233-236.

---

## Author Comment (AC1) · 13 Sep 2017

**Response to Reviewer #1**

**We thank the reviewer for a careful and thorough assessment of our manuscript. Below, we provide a point-by-point response to the comments.**

Dear Editor,

I have now read the manuscript by Frieling et al. entitled "Tropical Atlantic Climate and Ecosystem Regime Shifts during the Paleocene-Eocene Thermal Maximum". The manuscript presents new dinocyst and sedimentological data from the Paleocene/Eocene thermal maximum as recorded at tropical Site 959. The present paper is companion of another paper on the same record submitted somewhere else, and in its main conclusion follows up what previously found by Frieling et al. (2017) at a tropical shallow marine section deposited in Nigeria. In this paper the authors suggest that extreme warming during the PETM within this shallow and possibly restricted basin knocked off planktonic eukaryotic productivity. In the manuscript under consideration they came to the same conclusion, arguing for a collapse of marine eukaryotic productivity due to extreme temperatures also at the fully open ocean setting of Site 959. This represents the main finding of the paper. This is an interesting finding as it highlights how in the open ocean temperature alone may threats pelagic food webs, especially at low latitudes. On the other hand, the paper in my opinion suffers from the fact that several important aspects concerning the record of the PETM at the studied site are shown but not treated in this manuscript. The reader throughout the text is referred to the companion paper that is still – at least to my knowledge – under revision/submitted somewhere else. Since the PETM from Site 959 has never been described before, this in my opinion hampers a full overview of the environmental/oceanic conditions at Site 959 and especially on how the PETM was expressed at this locality, and somehow weaken the author main findings. The sedimentary and isotopic records of the PETM at Site 959 are quite peculiar, and to my knowledge are not alike anyone else found. However what this might mean in term of depositional environment, water column conditions, and in relation with the biotic data presented is insufficiently discussed here, and the reader is referred to the companion paper. I think this paper could be published after some substantial revisions. However I also believe this paper would acquire more strength and relevance if published once the companion paper will be published, when the authors will be free to openly discuss their other findings.

**Author response:**

The reviewer points out that the manuscript currently suffers from a lack of background information, which is to be published elsewhere (referenced here as Frieling et al., submitted). This paper was first submitted early this year but it has suffered from an unexpectedly large delay. We see options to alleviate the reviewer's concerns without reiterating information in this manuscript. We would be more than happy to share the submitted paper with the editor and reviewers, or as the reviewer suggests, publication of the current manuscript may be halted until publication of the first submitted manuscript. We will await editorial advice on this issue.

Line by line remarks:
Abstract:
Lines 23-25: **"The early stages of the PETM are marked by a typical acme of the tropical genus Apectodinium, which reaches abundances of up to 95%. Subsequently, dinocyst abundances diminish greatly, as do carbonate and pyritized silicate microfossils"** Dinocyst absolute abundance drops already within the porcellanite layer coinciding with the onset of the CIE at Site 959. Why you do not consider it at all? This should be explained and discussed in the text.

**Author response:**

We agree with the reviewer this had to be clarified and in our revised text we argue this initial drop in dinocyst numbers is likely due to the very organic-lean (oxidized) nature of the porcellanite. It should be noted that we interpret the porcellanite layer to be of turbiditic origin (p. 4 line 3-5 in original manuscript) and it follows that its deposition was extremely rapid. In Frieling et al. (submitted) we argue that organic matter is mixed in from above due to bioturbation. We now include these findings into this manuscript to provide the necessary background information.

Introduction:
Lines 10-13 pag 2: **"The CIE has a distinct shape; a rapid "onset" (1-5 kyr; (Kirtland Turner and Ridgwell, 2016; Zeebe et al., 2016) followed by a prolonged (50-70 kyr) period, the "body", of stable low 13C values and a recovery that lasts 42 - 100 kyr (Röhl et al., 2007; Abdul Aziz et al.,**

**2008; Murphy et al., 2010) to values that remain slightly 13C-depleted (0.5-1 ‰) relative to the latest Paleocene".**
Not correct. The cited papers estimate a duration of the CIE "body" of 70-100 kyr and of 100-130 kyr for the CIE recovery interval, please amend.

**Author response:**
Fixed.

Line 19 pag 2: More reference needed here as the quoted paper is about a tropical locality.

**Author response:**
The referenced paper (Frieling et al., 2017) is not solely about a tropical locality, but also contains a state-of-the-art global data compilation of sea surface temperature (SST) data from PETM sites that is accompanied by a thorough point-by-point comparison to fully-coupled climate model simulations. To our knowledge, the referenced paper is the only work that clearly shows the persistence of extra-tropical amplification during the PETM.

Line 20 pag 2: A widespread expansion of suboxic to anoxic waters during the PETM has been suggested by several works. Some more should be mentioned here, e.g. particularly relevant are: Nicolo et al., 2010; Zhou et al., 2014; 2016; Stassen et al., 2015.

**Author response:**
The reviewer points out that other works have also suggested a global expansion of suboxic to anoxic waters, which is correct. However, the referenced paper by Dickson et al. (2012) is the only that may show truly global deoxygenation as it is a function of Mo-isotope burial averaged across the ocean. Even considering excellent spatial coverage, site-specific oxygenation records always contain a degree of additional local effects. One could therefore argue that several dozens of other papers support the Mo-isotope record, but in our opinion this does not warrant referencing those works here.

Line 21 pag 2: See also Giusberti et al. 2016.

**Author response:**
We added this reference.

Line 24 pag 2: Please add some more recent work (a number of paper is likely to have been published on this topic since 2014).

**Author response:**
This regards a very general statement that requires citation of a review paper. Although it was published already a few years ago, the cited paper includes a review that still represents the most complete overview of environmental changes during the PETM and thus covers the statement made here.

Material

Lines 23-25 pag 3: A more in depth description of how the PETM is identified at Site 959 and how it compares to other records should be part of this paper as well. It does not suffice to refer to the companion paper submitted.

**Author response:**
As noted in the original manuscript (page 3 lines 21-23) the interval containing the PETM was originally identified based on calcareous nanofossil evidence (Shafik et al., 1998). To accommodate this comment by the reviewer, we now also add the magnitude of the carbon isotope excursion and point out variable carbon sourcing plays a role within the turbiditic porcellanite.

Lines 3-5 pag 4: **"The organic-lean biogenic silica and siliciclastics (804.1 – 803.8 mbsf) are likely derived from an allochthonous, potentially turbiditic, deposit, as reflected by anomalous Ti/Al ratios (Frieling et al., submitted), and obscures the exact onset of the carbon isotope excursion (CIE)".**

Why the authors argue that this layer has an allochthonous origin? Ti/Al per se does not indicate reworked material. To the opposite a number of evidence would suggest an autochthonous origin (gradually decreasing **δ**13C values, the drop in dinocyst abundance, a peak in pirityzed microfossils) possibly linked to the peculiar (extremely hot? euxinic?) seafloor and water column conditions at the onset of the PETM. This layer coincides with the onset of the CIE at Site 959, therefore its nature and possible origin and relation with the PETM should be better explained and discussed.

**Author response, see below:**

Lines 7-9 pag 4: **"The gradual decrease from ~ -27‰ at 804.09 mbsf to -30‰ at 803.8 mbsf is interpreted as a mixing line between organic matter produced during the Paleocene and the PETM (Frieling et al., submitted)".**
Mixing of pre-PETM and PETM organic matter would produce a flat line within the porcellanite layer, with isotopic values intermediate between those of the upper Paleocene and the early Eocene. In fact, the observed continuous decrease in carbon isotope values suggests the original onset of the CIE (or at least part of it) was being recorded. The CIE onset at Site 959 is somewhat similar to the onset of the CIE as recorded at Wilson Lake, New Jersey, see Stassen et al. 2015. This possibility should be explored, and the authors should better support their argument. If the porcellanite layer does record the onset of the CIE, this would imply an autochthonous origin for the layer. Why the authors so sharply disregard the possibility of an autochthonous origin?

**Author response (for both comments above):**
We include a substantial discussion on this matter in Frieling et al. (submitted), but we completely agree with the reviewer that this is difficult to interpret based solely on the evidence presented in this paper. In the revised version of this paper, we therefore explain that Frieling et al. (submitted) show that the gradual decrease can be replicated by a simple sediment mixing model that assumes bioturbation of PETM organic matter down into the organic-lean porcellanite and uses TOC(wt%) as a proxy for this process. This does not produce a flat line as the reviewer suggests, because mixing is never perfect and the presence of PETM organic matter in the porcellanite is accordingly variable. This scenario is consistent with bioturbation patterns in the core. Importantly, this simple model would not work if $\delta^{13}$C continued to change during the deposition of the porcellanite.
More fundamentally, we see no reason why the onset of the CIE should be gradual at Site 959, since no truly unambiguous (single-specimen) intermediate $\delta^{13}$C values have been recorded for the PETM. Indeed, most records with a gradual onset are evidently result of mixing.

Methods
The methods section must include a description of the age model mentioned in the discussion. Changes in accumulation rates affect microfossil absolute numbers, so an age model constraining accumulation rates must be described and be part of this work.

**Author response:**
We agree with the reviewer and will therefore include the required statements regarding sediment accumulation rates during the PETM. A more detailed discussion of the age model is presented in Frieling et al. (submitted).

Furthermore, this section must include how dinocyst absolute numbers were calculated.

**Author response:**
A description of how absolute dinocyst numbers are calculated was presented on p.4 line 27-28. We do not see how the use of this standard procedure (as described in Stockmarr, 1971) can be further clarified.

Also, the section lacks a description of the organic geochemistry methods used for lipid extraction and of the calibrations used to convert TEX86 into temperature. Since TEX86 data are discussed, the methods should provide this information as well.

**Author response, see below:**

Results

The results section should include a description of TEX86 data and the BIT index must be shown together with TEX86. The BIT index and its meaning in relation with TEX86 should then be discussed in the Discussion paragraph.

**Author response (for both comments above):**
The reviewer points out that a methods and results section is missing for the organic geochemical analyses in this manuscript. As stated in p. 3 line 6-7 these data are to be published in Frieling et al. (submitted) and we therefore do not include a methods or results section here.

Line 17 pag 6: **"The onset of the PETM is marked by an acme of Apectodinium…"**
In the result description the authors should refer to their (local) CIE signal rather than to the PETM, and they should always clearly keep this distinction. There should then be a section in the discussion in which the authors explain how they correlate the CIE at Site 959 to the PETM.

**Author response:**
The CIE has already been identified as representing the PETM based on calcareous nannofossil biostratigraphy, as stated in the materials section, so we do not quite understand this comment by the reviewer.

Line 18 pag 6: Not only the body of the CIE yields low abundances of dinocyst, the drop is firstly observed within the porcellanite layer. The author must explain why this is not taken into account.

**Author response:**
We now further clarify in the materials section that the porcellanite is regarded as a turbiditic deposit, with organic carbon mixed in from above. This causes low absolute numbers of dinoflagellate cysts deeper in the porcellanite, which was originally most likely deposited without any preserved dinocysts.

Line 22 pag 6: There should be "across" instead of "during".

**Author response:**
We will rephrase to 'within'.

Discussion:
Lines 28-29 pag 7: **"This observation is important since we can, although with caution, use these as indicators of environmental change that is not associated with background cyclic variability".**
The authors should further explore this.

**Author response:**
We now further clarify the implications of this statement.

Line 21-23 pag 9: **"The high percentages of Apectodinium in the porcellanite (804.1 - 803.9mbsf) are most likely mixed in from above (803.85 - 803.75 mbsf), similar to organic matter (Frieling et al., submitted), since assemblages and species are very similar"**
Again, why it could not be an original signal? Abundance peaks of **Apectodinium** are recorded almost everywhere at the onset of the PETM.

**Author response:**
We do not dispute the abundance of *Apectodinium* in the early stages of the PETM. However, we interpret the porcellanite to be of turbiditic origin with organic matter mixed in from above. Therefore, it is unlikely that these specimens are *in situ*.

Line 24 pag 9: paragraph title: **5.2.3 Microfossil decline and deoxygenation during peak PETM"**
What's the peak of the PETM at Site 959? The author never states what they interpret to be the peak PETM at the studied site. This should be part of the discussion I mentioned above in this revision.

**Author response:**
We agree with the reviewer that this should be stated more clearly. We now include a statement that we interpret the "body" phase as peak-PETM.

Lines 26-27 pag 9: Accumulation rates are mentioned but an age model is not described in the methods. It looks like the period misses a verb.

**Author response:**
This will be clarified and included in the materials section.

Lines 2-4 pag 11: **"No such short-term extreme biotic variation is known for any PETM site across the globe, certainly not for the recovery interval, and indicates highly variable environmental conditions."**
This is very poorly phrased and can generate confusion. Short term extreme biotic variations are widespread observed, and they represent one of the peculiar features of the PETM as well as one of the reasons why it is so much studied. Think only to the abrupt benthic foraminiferal extinction event at the onset of the PETM. Besides, short term biotic variation across the recovery phase have been observed also at other localities, in particular at shallow/marginal settings, e.g. see Luciani et al., 2007; Stassen et al., 2015; Giusberti et al., 2016.

**Author response:**
The reviewer's comment shows our text may cause some confusion between the high-amplitude variations at Site 959 and low amplitude variations observed in other high-resolution studies. The amplitude and perhaps also the timescales of the observed variation are far more extreme than observed in any of the records referenced by the reviewer. Dinocyst assemblages at Site 959 are dominated by different species on cm/(sub-)millennial scales during the recovery. We will properly rephrase to optimally clarify this point in the revised text.

Lines 10-11 pag 12: **"The combined information suggests that eukaryote activity in the mixed layer was suppressed not only in Nigeria (Frieling et al., 2017) but also at the more offshore Site 959. Similar to Nigeria, there is hence no evidence that the low numbers of dinocysts resulted from**
**severe stratification and anoxia…"**
In a previous paragraph the authors state: **"At Site 959, we find strong indications of decreased oxygen concentrations in bottom waters in the form of increased organic matter burial fluxes, as assessed through reconstructed accumulation rates and TOCwt%. Moreover, increasing Corg/Ptot ratios (Fig. 3i) relate to preferential regeneration of phosphorus from sediments under anoxic conditions (Slomp et al., 2002; Algeo and Ingall, 2007)….. The combined information from Site 959 and Nigeria suggests that oxygen minimum zones during the PETM expanded upwards onto the shelf and downwards to the paleodepth of Site 959 (>1000 m) in the eastern tropical Atlantic, a phenomenon very similar to modern trends (e.g., Stramma et al., 2008). ".**

Please make your data interpretation consistent.

**Author response:**
The interpretations are fully consistent. The severity of anoxia at Site 959 was less compared to the shelf section. At Site 959 we find no evidence for photic zone or bottom water euxinia. More importantly, and similar to the data presented in Frieling et al. (2017), the deoxygenation was asynchronous with the demise of eukaryotes and must therefore be decoupled. We have clarified this in the revised manuscript.

Typing Errors:
Fig. 3: Core sections should be named 42R-2, 42R-1 etc instead of 42X-2 etc
Line 6 pag 7: "a" missing
Line 1 pag 9: "the" genus **Apectodinium.**

**Author response:**
Fixed.

---

## Author Comment (AC2) · 13 Sep 2017

**Response to Reviewer #2**

We thank the reviewer for a critical but very constructive review of our manuscript that has forced us to carefully review established concepts within dinocyst paleoecology in the light of the new data presented in our manuscript. As advised by the reviewer, we will include an overview of potential heat-stress evidence during the PETM in our revised manuscript. Below, we provide a point-by-point response to the comments.

This contribution provides detailed new dinoflagellate assemblage data through the PETM from the western tropical Atlantic, supported by additional new bulk sediment chemistry and magnetic susceptibility measurements. Overall it is clearly presented, well written and a solid contribution to the dataset of surface ecosystem responses to the PETM warming event. I do have a couple of concerns and some minor comments that need to be addressed by the authors before publication.

1. One of the key points made within this paper is that, to quote from the abstract: "The combined paleoenvironmental information from Site 959 and a close by shelf site in Nigeria implies the general absence of eukaryotic surface-dwelling microplankton during peak PETM warmth is most likely caused by heat stress." My concern is that evidence presented from selected sites is framed to make inferences about global responses and environmental drivers: "Site 959 and a close by shelf site in Nigeria: : :. implies the general absence of: : :". Within a few words they've gone from local and specific ("close by") to a "general absence". This is a problem because this group of authors are foremost in the analysis of PETM dinoflagellate (and other) records. The quality of their regular outputs, and regard within the community, gives them a very strong influence on shaping the accepted narrative and interpretation of data. With this is mind, I think they have to be exceptionally careful about the claims that are made and that these fully take into account uncertainty in going from the observed data to interpretation.

**Author response:**
We thank the reviewer for being critical on this and we fully agree we need to be careful regarding claims. We did not intend to claim that eukaryotes died off in the entire tropical band. 'General absence' was not intended to suggest that, but rather to reflect the equatorial eastern Atlantic region, notably the study area covering Site 959 and the Nigerian realm we studied for our 2017 paper. We will specify this in our revised manuscript.

In this case they may well be correct and are presenting a substantive account of the true ecosystem responses, but my concern is that only references that support this "heat stress" and tropical exclusion of eukaryotes are cited – self-citations and Aze et al. (2014) and Yamaguchi & Norris (2015). A stronger case would include a wider overview and consideration of tropical sites where there is less or no evidence for the exclusion of eukaryotes.

**Author response:**
We agree and will include a section in the revised paper to discuss this issue. We stress that the combined evidence from Site 959 and Nigeria is quite unique because there is independent information on (excessively high) temperature and other environmental parameters from two very different but close by geological settings. Only this allows extracting the SST the dominant forcing factor. For most other sites, this is impossible to accomplish. Indeed, dissolution of carbonates and overall low-organic carbon content of sediments hampers tracking biotic change and how this relates to various environmental factors for the open ocean equatorial and tropical (20 ºS – 20 ºN) sites (e.g. ODP Site 999, 1001, 1220, 1221, 1258, 1260). The records from Site 865 and 1209 do not show a planktic eukaryote demise, but it should be noted that SST reconstructions from these sites do not show the excessively high temperatures (>35ºC) recorded at Site 959, Nigeria or Tanzania and that these SST trends are consistent with model results (Frieling et al., 2017). The southern Tethyan shelf (Egypt) may also be considered shallow enough to avoid carbonate dissolution through CCD rise. However, the PETM in Egypt is marked by black shales devoid of carbonates and yet, as Speijer and Wagner, (2002) note, contain no dinocysts, spores or pollen. These observations could be interpreted as supportive evidence for heat-stress among planktic eukaryotes, albeit less confidently. This will be discussed in the new section.

For example, the Tanzanian section discussed by Aze et al. (2014) also has records of coccolithophore communities and calcareous dinoflagellates throughout the PETM – calcareous dinoflagellates are

actually shown to increase in abundance during the PETM (Bown and Pearson, 2009).

**Author response:**

This is a nice observation. It should be noted that calcareous dinoflagellates have very different ecologies than organic cyst producing dinoflagellates. Indeed, the calcareous dinocysts increase in relative abundance – but remain present in very low absolute abundance as do coccolithophores. One speculative interpretation of these data would be that also these organisms were outcompeted by the more heat-stress resilient prokaryotes. This will be included in the additional section in the revised paper.

Similar records of persistence of coccolithophore communities and increase in calc. dinos are shown from the tropical Pacific, ODP Site 1209 (Gibbs et al., 2006b). In Site ODP 1209 there is an increase in phytoplankton turnover (Gibbs et al., 2006a), which may be related to heat stress, but there is little evidence for a total exclusion of eukaryotic microplankton from this tropical location.

**Author response:**

As the reviewer rightfully points out, the abundances of coccolithophores and calcareous dinocysts are unaffected by any heat-stress at IODP Site 1209. However, Site 1209 may not be as hot as Site 959, Nigeria or Tanzania during the PETM. Maximum recorded Mg/Ca SST estimates for Site 1209 are 32-33 ºC, equivalent to those in the latest Paleocene in Nigeria (Frieling et al., 2017). Also this evidence will be included in the additional section in the revised paper.

There may be reasons for this increase in calc. dinos. in both the Tanzanian and Pacific tropical sites, and this might support some of this groups' interpretations, but there needs to be some recognition that these other records exist and then an integration of data to form a more solid interpretation of the wider (/global) patterns of change. In this instance, is there a case for any ecological exclusion of dinoflagellates be limited to the (eastern) equatorial Atlantic? I don't think there is strong evidence (yet) to extrapolate from these two relatively close sites (Nigeria and ODP 959) to a global response in the tropical oceans. Any associated sea surface temperature records from these locations might also just represent localized effects that aren't replicated in either the tropical Pacific or Indian Oceans.

**Author response:**

We agree and did not intend to suggest otherwise. We will make sure to adapt the text accordingly to argue that only the hottest parts of the ocean may have been affected. As indicated in Frieling et al. (2017), the studied region was likely one of those based on the climate model simulations.

2. The use and referencing of a submitted manuscript "Frieling et al. submitted" is frustrating. This was not provided to reviewers. Although I don't think the conclusions of this manuscript rely on what may be contained within this other submission, one feels that we're being asked to review this paper with 20% of the interpretation (and data?) hidden from view. Ideally, I would rather this manuscript was not published until either the "submitted" manuscript was published or made available for reviewers and editors of this submission. For example, key interpretation of the CIE, its onset and the temperature data are all likely contained in this other submission. I would recommend that the editor at least be able to see this other submitted manuscript in confidence prior to any final publication of this paper, so that they can judge the degree of overlap.

**Author response:**

Reviewer #1 also raised this comment. The companion paper ("Frieling et al., submitted") was submitted early 2017 and we did not anticipate a delay of this magnitude. We would be more than happy to share the submitted paper with the editor and reviewers or publication of the current manuscript may be halted until publication of the first submitted manuscript. We will await editorial advice on this issue. In addition, we will replicate the needed information in this manuscript, with proper citation.

3. Related to the development of a narrative for PETM dinoflagellate records presented by this group over a number of years, I'm intrigued by the interpretation presented of changes in abundance of key indicator species that previously have been used to infer sea level change through the PETM in shelf sites (page 8).

*"From 804.4 mbsf, we find an increase in abundance of dinocysts belonging to, or closely related to the genus Areoligera (Areoligera complex sensu Sluijs and Brinkhuis, 2009). A relative abundance*

*increase of this genus was previously interpreted to reflect sea level rise at several shelf and slope sites during the PETM (Sluijs et al., 2008). However, Site 959 is located in an open ocean setting, which means water depth and shore proximity proportionally do not change as much as may be expected from sites on the continental shelf, especially if estimates of the amplitude of sea level rise across the onset of the PETM (5-20m, e.g., (Speijer and Morsi, 2002; Sluijs et al., 2008) are considered. The increase in Areoligera is further associated with a decrease in Spiniferites, consistent with other PETM records (e.g., Sluijs et al., 2008), including a recently published record from Nigeria (Frieling et al., 2017). Since we cannot distinguish between transported and local signals, we may either record a signal that is transported off the shelf, or a local signal that is similar to, but not related to sea level."*

I find this a little odd. If the dinoflagellate records are so subject to transport across shelf to the slope and deep ocean, what use are they in reconstructing relative position, from the marginal to oceanic? Which I thought was a substantial component of dinoflagellate paleoenvironmental interpretations? The other option presented is that this assemblage change is: "similar to, but not related to sea level." This seems more likely than pervasive long distance transport. But if there is an alternate environmental cause of this assemblage change in the open ocean sites, then doesn't this also somewhat question whether the interpretation - of the same assemblage changes through the PETM from shelf-records - as being caused by sea-level is open to some reinterpretation? Could there rather be a broader dinoflagellate assemblage change (increase in Areoligera) that is rather related to the wider environmental changes in the tropical / sub-tropical Atlantic and less controlled by sea level? If there are such major PETM environmental changes in the tropics / sub-tropics, such as the heat stress the authors propose plus potential changes in stratification and nutrient supply, wouldn't these be more likely the drivers of dinoflagellate assemblage changes than a relatively modest change in sea level? If so, then this seems like an appropriate place to put the previous interpretations into this new context for the non-expert reader. Again, I'd emphasize, that when this group of authors dominate the generation of PETM dinoflagellate records and the interpretation of them, it's also their responsibility to the external readership to directly address such questions as new data and interpretations arise.

**Author response:**
We thank the reviewer for being critical about this section, it incompletely described previous work and therefore did not make full sense. We therefore include a more substantial discussion on this matter in our revised manuscript, based on the following arguments.
The main challenge here is that the Site 959 record is the first PETM organic-walled dinocyst record from the open ocean. Generally, organic cyst-forming dinoflagellates are bound to the continental margin because upon production dinoflagellate cysts sinks to the ocean floor, where it spends a benthic stage of the life cycle (Fensome et al., 1996). After excystment, the dinoflagellate swims up to the sea surface but in open ocean settings, this is impossible. Therefore, only a few species are capable of sustaining in the open ocean (Zonneveld et al., 2013). What we omitted in the paper is to describe how the dinocyst assemblages do reflect this offshore setting, notably with the very high abundances of *Spiniferites*, which is a genus that is relatively more abundant with increasing distance to coastlines in the Paleogene and the modern (e.g., (Brinkhuis, 1994; Pross and Brinkhuis, 2005; Zonneveld et al., 2013). In such a setting, we would not necessarily expect common to abundant *Areoligera*, because on the shelf, *Areoligera* has been proven to be related to relatively near-shore, high-energy conditions based on independent information on sea level, grain size and relative contributions of terrestrial organic sedimentary components (e.g., Sluijs et al., 2006, 2008a, 2008b; Sluijs and Brinkhuis, 2009). In our view, therefore, the interpretations regarding the effect of sea level on the relative abundance of *Areoligera* during the PETM are solid.
It should be noted, however, that the PETM is associated with a drop in *Areoligera* abundance and a concomitant rise in *Spiniferites* in the relatively offshore locations in New Jersey, such as Bass River (Sluijs & Brinkhuis, 2009), and the Tawanui slope section in New Zealand (Crouch and Brinkhuis, 2005). In the pro-delta settings of Site 1172 on the East Tasman Plateau (Sluijs et al., 2011), the ACEX core in the Arctic Ocean (Sluijs et al., 2008a) and Spitsbergen (Harding et al., 2011), the PETM sees an influx of *Areoligera* into Paleocene freshwater dominated dinocyst assemblages, evidencing a more marine setting. In the nearby shelf site in Nigeria, *Areoligera* is a common constituent of the assemblage only directly before the PETM (Frieling et al., 2017), perhaps recording both the eustatic rise at the PETM and a latest Paleocene regression (Speijer and Morsi, 2002). With this overview, we lay out there is no global increase in *Areoligera*, but rather that the signal depends on the location of the site on the shelf, relative to the coastline and river outflows and the distal end of the margin.

This does leave the question open how to explain *Areoligera* abundances at Site 959 and this is not an

easy one given the limited available information from open ocean sites. We concur with the reviewer that long distance offshore transport is unlikely, particularly because we do not find accompanying abundances of terrestrial organic components (branched GDGTs, pollen and spores). We however note that there is a concomitant increase in the relative abundance of Goniodomids, which are typically associated with warm (seasonally) stratified waters in the Paleogene (e.g. Sluijs et al., 2011, 2014) and modern ocean (Zonneveld et al., 2013). It is inferred from both model results and carbonate proxy data there is a strong but shallow permanent thermocline (and this likely also pertains to the pycnocline) in this region (Frieling et al., 2017), which could act as a substitute seafloor to cyst-forming dinoflagellates, similar to recorded during phases of strong stratification in the Arabian Sea during the last glacial cycle (Reichart et al., 2004). We therefore speculate that the higher percentages of *Areoligera* here may be facilitated by strong(er) stratification in the latest Paleocene, rather than to sea level change. The waters of above the thermocline may emulate the high-energy environment *Areoligera* prefers.

Other comments:
1. Given that Thomas Westerhold is a co-author, I'm surprised that there is no mention, use or citation of the latest age model assessment for the PETM: Westerhold et al. 2017. Clim. Past Discuss. https://doi.org/10.5194/cp-2017-74. And specifically the durations provided for the PETM in this paper appear to be at odds with Westerhold et al. 2017.

**Author response:**
The revised text is made consistent with all available literature, including this paper that is still in review.

2. Use of capitalization for informal sub-epochs / sub-series: e.g. Page 2, line 3: "during the Late Paleocene and Early Eocene: : :". See Pearson et al. 2017. Episodes, http://dx.doi.org/10.18814/epiiugs/2017/v40i1/017002

**Author response:**
Fixed.

Bown, P., and Pearson, P., 2009, Calcareous plankton evolution and the Paleocene/Eocene thermal maximum event: New evidence from Tanzania: Marine Micropaleontology, v. 71, no. 1-2, p. 60-70.

Gibbs, S., Bown, P., Sessa, J., Bralower, T., and Wilson, P., 2006a, Nannoplankton extinction and origination across the Paleocene-Eocene Thermal Maximum: Science, v. 314, no. 5806, p. 1770.

Gibbs, S., Bralower, T., Bown, P., Zachos, J., and Bybell, L., 2006b, Shelf and openocean calcareous phytoplankton assemblages across the Paleocene-Eocene Thermal Maximum: Implications for global productivity gradients: Geology, v. 34, no. 4, p. 233-236.

**References used in author response**

[revised manuscript text omitted]